



# Exploring Long-term Monthly Prediction of Precipitation Isotopes over Southeast Asia: A Comparative Analysis of Machine-Learning Models

Mojtaba Heydarizad[1], Liu Zhongfang[1*], Nathsuda Pumijumnong[2], Masoud Minaei[3,4], Pouya Salari[5], Rogert Sorí [6], Hamid Ghalibaf Mohammadabadi [7]

[1]State Key Laboratory of Marine Geology, Tongji University, Shanghai, 200092, China

[2] Faculty of Environment and Resource Studies, Mahidol University, Nakhon Pathom, 73170, Thailand

[3] Department of Geography, Ferdowsi University of Mashhad, Mashhad 917794883, Iran

[4] Geographic Information Science/System and Remote Sensing Laboratory (GISSRS: Lab), Ferdowsi University of Mashhad, Mashhad 9177794883, Iran

[5] Department of Geology, Ferdowsi University of Mashhad, Mashhad 917751436, Iran

[6] Centro de Investigación Mariña, Universidade de Vigo, Environmental Physics Laboratory (EPhysLab), Campus As Lagoas s/n, Ourense, 32004, Spain

[7] Department of Computer Engineering, Ferdowsi University of Mashhad, Mashhad 9177948974, Iran

*Correspondence to*: Liu Zhongfang ( liuzf406@tongji.edu.cn)

**Abstract.** Using stable isotope methods is essential for studying tropical hydrology and climatology. The purpose of this research was to investigate the influence of large-scale climate modes (teleconnection indices) and local meteorological parameters on the stable isotope contents in six different stations, including Bangkok, Kuala Lumpur, Jakarta, Kota Bharu, Jayapura, and Singapore in Southeast Asia. To achieve this goal, several machine learning (ML) techniques were employed, such as shallow neural network (SNN), deep neural network (DNN), decision tree (DT), random forest (RF), and extreme gradient boosting (XGBoost). XGBoost demonstrated the highest accuracy across the majority of studied stations, with a $R^2$ = 0.91, VNS=0.90, AIC= 405, BIC=410, and RMSE = 0.76. Additionally, DNN exhibited superior accuracy in specific cases, achieving a $R^2$ = 0.87, VNS=0.87, AIC = 445, BIC = 460, and RMSE = 1.10. Furthermore, a bootstrap analysis was conducted to assess the uncertainty of the simulated data in each station. The results of this analysis demonstrated acceptable accuracy, as the majority of simulated data points fell within the 95% confidence intervals.Finally, stable isotope contents in precipitation were forecasted for one year using Vector Autoregression (VAR) and ML techniques. This study underscores the efficacy of ML techniques in both simulating and forecasting stable isotope contents with high precision. The inclusion of specific accuracy metrics strengthens the validity of claims in this study and provides a clearer picture of the quantitative outcomes of this research.





Keywords: Precipitation isotopes, Southeast Asia, Prediction, Machine-learning models, Bootstrap uncertainty analysis, Validation

## 1 Introduction

Precipitation is the most essential part of the water cycle, which has dominant role in hydrological and climatological systems (Porntepkasemsan et al., 2016). Hence, studying precipitation with accurate proxies such as stable isotopes ($\delta\,^{18}$O and $\delta\,^2$H) can help obtain invaluable information regarding the water cycle and climatic changes in a study region. Since the discovery of the strong correlation between $^{18}$O and $^2$H in water by Harmon Craig (1961), numerous surveys on stable water isotopes have been conducted to investigate hydrological characteristics at global and regional scales (Clark and Fritz,

1997). In addition, a global network of isotopes in precipitation (GNIP) was established for hydroclimate studies with the help of WMO and the IAEA. Some of the GNIP stations were operational for short periods or even just one year, while some others, for example, Bangkok, Ottawa, Tehran, etc. were active for more than 30 years. These long-term records of precipitation isotopes have offered valuable information about regional and global hydrological and climatic processes (IAEA/GNIP, 2018).

Although precipitation isotopes have been widely applied in numerous hydroclimate investigations, they are subject to some disadvantages and shortcomings. The most crucial shortcoming is the high expense of developing and operating a precipitation sampling network for stable isotope measurements. In addition, precipitation sampling is not always feasible in some remote areas, particularly in hard-to-reach regions. These concerns point to the need for simulations that allow the estimation of precipitation isotopes based on existing data sets. To simulate $\delta^{18}$O and $\delta^2$H in precipitation, isotope-equipped

general circulation models (GCMs) are powerful tools. However, these numerical models are challenging due to the complexity of the physical processes involved and their high computational cost. It also has been found that some numerical models fail to capture long-term data on precipitation isotopes (Kopec et al., 2015). In contrast, statistical models provide a simple, but effective, method for short-term precipitation isotope predictions by building relationships between isotopes and climate parameters. There are various statistical methods, such as the ridge, lasso, stepwise, and elastic net methods, that

have been used to predict precipitation isotopes (Mohammadzadeh et al., 2020; Mohammadzadeh and Heydarizad, 2019). In addition to these simple statistical models, machine learning (ML) techniques have been demonstrated to be remarkably successful in a variety of applications, including hydroclimate predictions. ML is a data analysis method that is a branch of artificial intelligence. ML techniques are based on the concept that systems can learn from raw data, recognize existing patterns, and make choices with minimal human interaction (Rahmati et al., 2017). The usage of ML started with the

application of artificial neural network (ANN) techniques (Banerjee et al., 2011; Barzegar and Asghari Moghadam, 2016) developed by McCulloch and Pitts in 1943 (Mcculloch and Pitts, 1943). Since then, numerous ML models have been developed and applied in different science fileds. Several ML methods, including the neural network (Banerjee et al., 2011; Cerar et al., 2018; Guzman et al., 2017; Mirarabi et al., 2019; Narayanan and Chintalapati, 2020; Sahour et al., 2020;



Wunsch et al., 2018), decision tree (Lee and Lee, 2015; Samadianfard et al., 2022; Xie et al., 2021), random forest (Kenda et
al., 2018; Koch et al., 2019; Wang et al., 2018), gradient-boosting (Malik et al., 2022; Ni et al., 2020; Song et al., 2022), and
extreme gradient-boosting (Narayanan and Chintalapati, 2020; Sahour et al., 2020) techniques, have been applied in
numerous hydrological studies. However, predictions about precipitation isotopes based on ML methods
have been rarely reported (Erdélyi et al., 2023; Heydarizad et al., 2023a; Nelson et al., 2021).

In this study, authors built on observational precipitation isotope data from Southeast Asia, using GNIP stations that are
located in a tropical climate and have long-term isotope records, and explored the predictive potential for monthly
precipitation isotopes using different ML methods. The authors first determined the relative importance of large-scale
climate indices and local meteorological parameters for influencing Southeast Asia precipitation isotopes using various ML
models. The authors then screened a subset of climate parameters as the best predictor variables for the different predictive
models. Finally, the authors evaluated the performance of these predictive models and chose the best-performing one for
precipitation isotope predictions.

## 2 Climatology of the study region

Southeast Asia is mainly dominated by tropical monsoon (Am) and, to a lesser extent, tropical savanna (Aw) climates,
according to the Köppen climate classification. The Am of Southeast Asia consists of two independent components: the
southwest (SW) monsoon and the northeast (NE) monsoon (Manisan, 1995) (Fig. 1a).

The SW monsoon starts in mid-May and ends in mid-October, causing significant precipitation events in Southeast Asia,
especially Thailand, from August to September (Khedari et al., 2002). During the SW monsoon season, Southeast Asia is
dominated by the influence of two main air masses. An air mass originating the Indian Ocean transports a large amount of
moisture into Southeast Asia (Nieuwolt, 1981), which couples with the unstable air mass emerging from the South Pacific
Ocean and Australia, resulting in more intense precipitation events. On the other hand, the NE monsoon prevails from mid-
October to the next April, during which most parts of Southeast Asia, particularly Thailand, are controlled by cold and dry
air masses from the Pacific Ocean (Nieuwolt, 1981). Between the two monsoons, there exists a period known as the inter-
monsoon phase, during which the air temperature increases significantly (Khedari et al., 2002).





**Figure 1** The NE and SW monsoon trajectories toward Southeast Asia and the GNIP/study stations location (a), and the monthly variation in air temperature and precipitation amount in the Southeast Asia region (data derived from GNIP station data sets) (b).

During the NE monsoon, the monthly precipitation and temperature demonstrate lower values compared to the annual average in Southeast Asia, and these parameters show the lowest values in December. In contrast, when the SW monsoon occurs, the monthly rainfall and air temperature exhibit greater values than the annual average. The highest monthly precipitation occurs in September during the SW monsoon, while the air temperature shows the highest values in the transition period (Fig. 1b).



Studying the wind speed and direction based on the NCEP/NCAR reanalysis (NOAA, 2020) from the NOAA at a pressure level of 850 hPa showed that strong winds mainly transfer moisture from the Indian Ocean toward Southeast Asia during the SW monsoon (Fig. 2a). However, during the NE monsoon, strong winds are observed from the northeastern and eastern directions toward Southeast Asia and transfer the moisture of the South China Sea to this region (Fig. 2b). During the inter-monsoon phase (Fig. 2c), the powerful winds seen during the SW and NE monsoon periods are not observed. This is the reason for the stable atmospheric conditions and negligible moisture transfer toward Southeast Asia during this period.

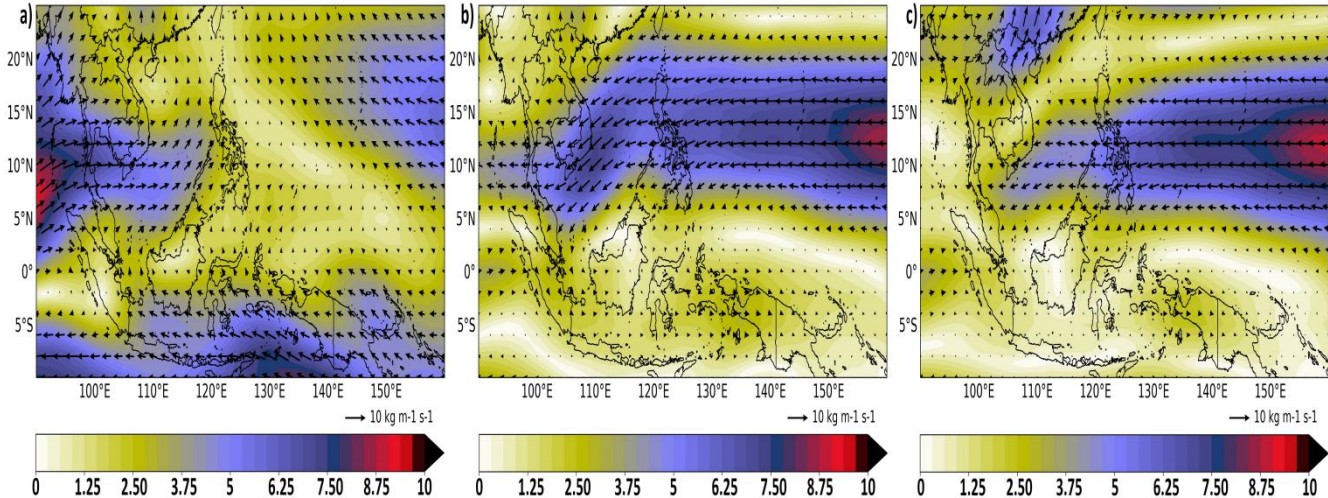

**Figure 2** Wind speed and direction maps during the SW (a) and NE (b) monsoons as well as the transition period (c) over Southeast Asia (0°–25°N, 90°–115°E).

On the other hand, studying the variations in the monthly precipitation distribution as well as the atmospheric stability (which is typically studied by calculating ω) at a 500 hPa pressure level showed negative values for ω, which represents atmospheric instability mainly over the southern, western, and northwestern parts of Southeast Asia during the SW monsoon (Fig. 3a). The daily precipitation amount also showed higher values in the regions with atmospheric instability (Fig. 3d) during the SW monsoon. During the NE monsoon, strong unstable atmospheric conditions were observed in the southern part of Southeast Asia including, Malaysia and Indonesia (Fig. 3b). This was followed by high precipitation amounts, mainly in the southern part of Southeast Asia (Fig. 3e). Finally, atmospheric instability exists over the southern and eastern parts of Southeast Asia during the inter-monsoon phase (Fig. 3c), followed by an increase in precipitation amount in these regions (Fig. 3f).



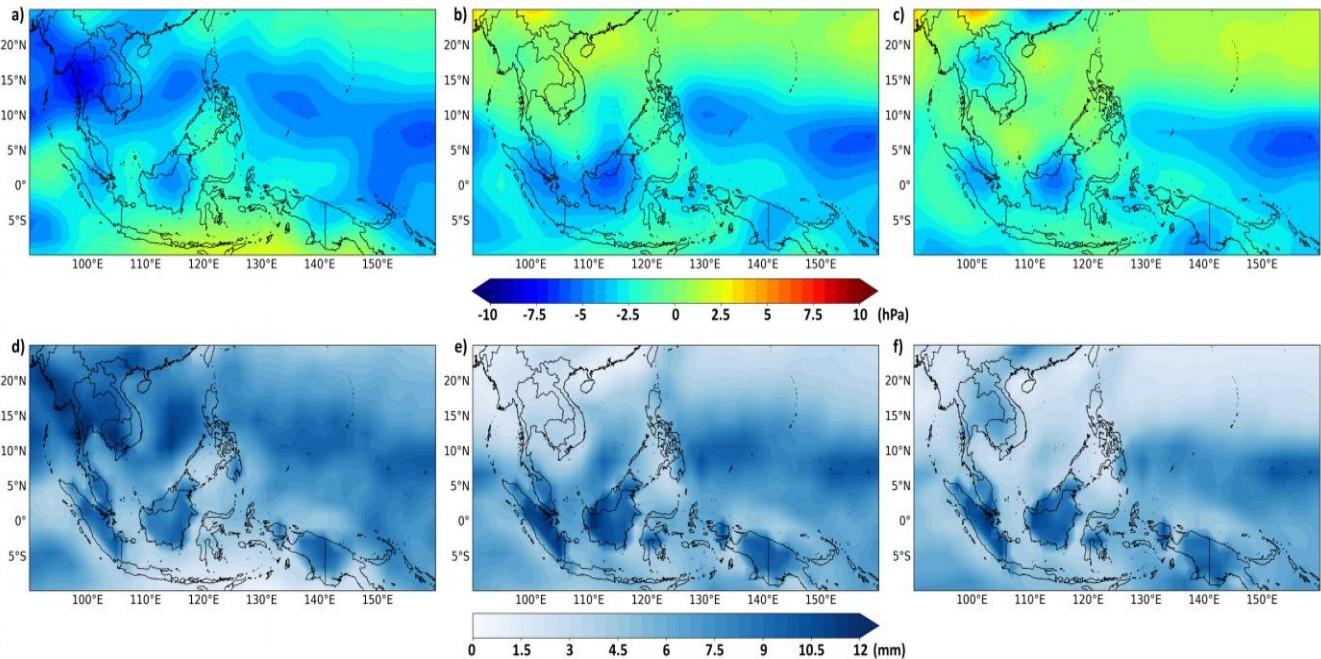


**Figure 3** The stability of atmospheric (Omega ω) variations (a,b,c) and precipitation amount distribution (d,e,f) over Southeast Asia (0°–25°N, 90°–115°E) for the SW and NE monsoons and the transition period, respectively. The data source is the NCEP NCAR reanalysis 1.

## 3 Materials and methods

During this survey, the stable isotope signatures in precipitation recorded by the GNIP at six different stations across Southeast Asia, including Bangkok, Kuala Lumpur, Jakarta, Kota Bharu, Jayapura, and Singapore, were investigated. The stable isotopes in precipitation were shown in δ relative to the VSMOW, and in ‰ units by Eq.(1):

$$\delta 18Osample = \left(\frac{\left(\frac{18O}{16O}\right) sample}{\left(\frac{18O}{16O}\right) reference} - 1\right)^{1} * 1000‰ \quad VSMOW \qquad (1)$$


The $^{18}$O and $^{2}$H isotopes had analytical uncertainties of 0.1 ‰ and 1‰, respectively. In this study, the authors omitted the stable isotope content in cases where the calculated deuterium excess (d-excess) value was higher than 50 ‰ or lower than -30 ‰. According to (Nelson et al., 2021), these stable isotopes lead to extreme precipitation events, which occur rarely in the monthly predictor timescale.

To simulate the stable isotope content (target variable) in precipitation of the studied stations, local variables (including the potential air evaporation, wind speed, vapor pressure, air temperature, relative humidity at 850 mb (the pressure level at which most of the moisture responsible for precipitation in this region originates), and precipitation amount) and regional variables (teleconnection indices) were independent variables.





The local parameters, including the potential air evaporation and wind speed, have been obtained from the NOAA website

(NOAA, 2018a). However, the vapor pressure, precipitation amount, and air temperature data were provided by the GNIP stations. According to previous studies (Ichiyanagi and Yamanaka, 2005; Pong et al., 2002), the leading teleconnection indices that influence the south of Asia and Thailand include the quasi-biennial oscillation (QBO), the Pacific decadal oscillation (PDO), the Madden–Julian oscillation (MJO), the bivariate ENSO (BEST), the Southern Oscillation Index (SOI), and the Indian Ocean dipole (IOD) time series. These are available on the NOAA website (NOAA, 2018a, 2018b) and were

used as independent variables (regional parameters) in this study.

Several prediction models using various packages in R were used to predict the stable isotope contents in precipitation. Initially artificial neural networks (ANNs), including shallow neural networks (SNNs) and deep neural networks (DNNs), were utilized. Unlike conventional statistical techniques such as regression methods, problems with complex nonlinear interactions are very well suited for neural networks (M.H and Darand, 2009; Mislan et al., 2015; Purnomo et al., 2017;

Schroeter, 2016).

Then, decision trees (DTs) and random forest (RF) ML techniques were used to predict the stable isotope contents. Finally, to achieve a more portable and accurate algorithm capable of omitting the computational limits observed in other ML models, the extreme gradient-boosting (XGboost) model was applied.

After constructing the model using training data, its precision is assessed by employing the ideal dataset. To authenticate the

ML techniques, a commonly utilized approach called cross-validation (v-fold variant) was implemented, utilizing the rsample package (Silge et al., 2022) in R language (R core team, 2018). The procedure includes spiliting the datasets into train and test sets. An essential aspect while splitting the data into these sets is to guarantee that the distribution of the test data accurately reflects the entire dataset (Frick et al., 2023). In v-fold cross-validation, the dataset is spilited to v separate and non overlapping subsets randomly. This division is done to create training and testing sets.

After completing the training and testing stages in each developed model, the precision of the model was evaluated using the coefficient of determination ($R^2$), the Nash Sutcliffe model efficiency coefficient (NSE), the root mean square error (RMSE), Akaike information criterion (AIC), and Bayesian information criterion (BIC) to determine the most accurate method for stable isotope simulation. $R^2$, NSE, and RMSE can indicate the degree to which a model accurately presents the data. In contrast, AIC and BIC can be used to compare various models, considering their level of hardness.

The reliability of the model's predictions and the accuracy of the simulated data were evaluated through a bootstrap uncertainty analysis, which considered multiple metrics. This enabled calculating the model's level of uncertainty and offered a comprehensive evaluation of its effectiveness.

In the final step, the stable isotope contents in precipitation were forecasted for one year at each station after the GNIP precipitation sampling project was terminated. To conduct the forecasting procedure, the most accurate ML model in each

station, as well as vector autoregression (VAR), were applied. The VAR model procedure starts by determining the number of folds for LOOCV (Leave One Out Cross Validation) and initializing a vector to store LOOCV outputs. It also initializes variables to store minimum CI value and iteration with minimum CI value. Then, it conducts LOOCV by iterating over the



number of folds defined earlier. In each iteration, it determines the index for the test set, defines the test set, defines the training set, determines optimal lag order using AIC (Akaike Information Criterion), fits the VAR model to the training set

with optimal lag order, makes a forecast for the test set, computes squared error for the test set and stores LOOCV outputs. Finally, the results of ML models were compared with the outputs of VAR models. To evaluate two models, firstly, the LOOCV procedure was used to estimate the performance of each model when they were used to make predictions on data not used to train the model. Then, the RMSE error was calculated for each model using the predicted values and measured values, and the model which demonstarted the lowest RMSE error was chosen as the most accurate.


## 4 Results and discussion

### 4.1 Choosing the best input parameters for building ML models

Choosing the optimal predictors for creating a simulation of the stable isotope contents of precipitation at the Southeast

Asian stations is the most essential step in each ML modeling. Eliminating irrelevant and redundant predictors will increase the robustness of the developed machine learning models while reducing computational expenses (Akbarian et al., 2023). Pearson correlation coefficients at a 95% confidence level were used to examine the main factors influencing stable isotopes in precipitation at the studied stations (Fig. 4).





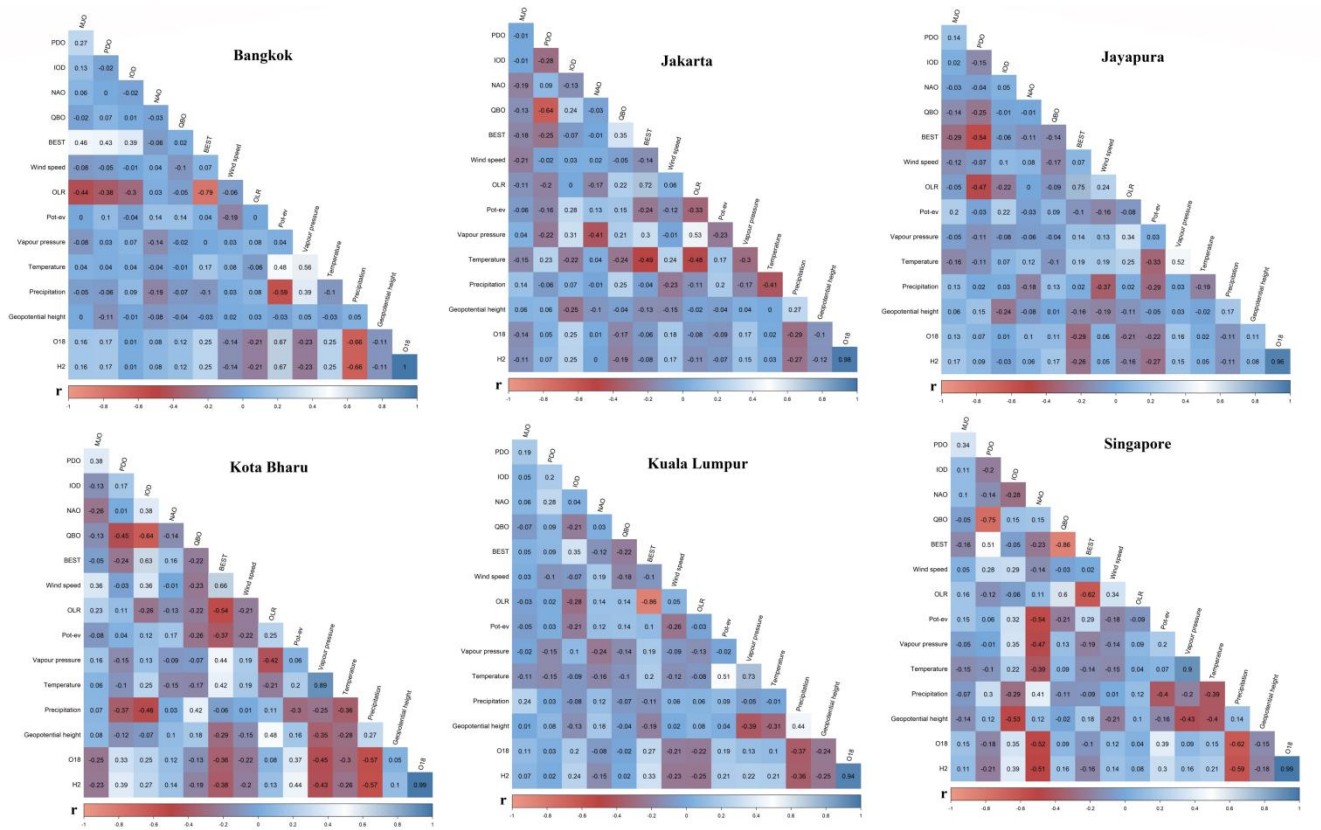

**Figure 4** (a) Pearson correlation coefficients and (b) Spearman's rank correlation were applied to examine the factors influencing the stable isotope composition of precipitation at the GNIP stations in Southeast Asia. An asterisk marks the pairs that have a significant difference in statistics (*).

This research discovered pairs with statistical significance (sig<0.05) between the teleconnection indices. In Jakarta station, QBO had a remarkable correlation with PDO (r=-0.64). In Kota Bharu, IOD had a correlation of r=-0.64 with QBO. In Singapore station, PDO correlated r=-0.75, and BEST correlated r=-0.86 with QBO. The effects of teleconnection indices on climatic parameters were also investigated. BEST had a notable correlation with OLR in most of the studied stations, including Bangkok (r=-0.79), Jakarta (r=0.72), Jayapura (r=0.75), Kuala Lumpur (r=-0.86), and Singapore (r=-0.62). However, QBO only strongly correlated with OLR (r=0.60) in Singapore station.

Among the local parameters, potential evaporation was found to correlate with temperature (r=0.51) in Kuala Lumpur station and with precipitation (r=-0.59) in Bangkok station. Vapor pressure also correlated with temperature in most stations, including Bangkok (r=0.56), Jayapura (r=0.52), Kota Bharu (r=0.89), Kuala Lumpur (r=0.73), and Singapore (r=0.90). The inverse relationship between the amount of precipitation and the potential evaporation showed that more moisture in the air and more precipitation per month usually lowered the potential evaporation (Clark and Fritz, 1997). On the contrary,





more vapor pressure in the atmosphere (which, together with atmospheric instability, is a key factor for precipitation to happen) led to more precipitation in the study sites. Moreover, the relationship between air temperature and vapor pressure also revealed that higher air temperature caused more surface water resources to evaporate, resulting in a substantial rise in

atmospheric vapor pressure (Thornthwaite, 1948).

The results demonstrated that precipitation had a significant influence on stable isotopes in precipitation. However, other parameters, such as teleconnection indices, had little impact on most of the stations. The stable isotope signatures were negatively correlated with precipitation, which can be attributed to the impact of precipitation amount. As the amount of precipitation increases, the heavier isotopes, such as $^{18}$O and $^{2}$H, preferentially condense and are removed from the vapor

(cloud), while the lighter isotopes remain in the vapor phase. This results in the progressive depletion of heavy isotopes in the remaining vapor as precipitation continues. Therefore, the stable isotope content in precipitation tends to decrease as precipitation increases (Clark and Fritz, 1997).

In addition to the Pearson correlation coefficient, the elimination by importance method has also been used at the studied stations for predictor selection. Several methods for selecting important predictors, such as Recursive Feature Elimination

(RFE) and Lasso regression have been used. In the RFE method, all possible combinations of predictors are used to run the models. The explanatory power of each predictor is determined by RFE, and predictors with lower importance criteria are eliminated by the models in each search step. In the RFE method used in this study, the random forest (RF) was used as the underlying model for feature selection. The main predictors were selected based on 10 fold cross validation (K fold method), and the RMSE method was used to evaluate the model's performance during feature selection. In addition to the RFE

method, the Lasso regression method has also been used to determine the most important predictors. This method performs both variable selection and regularization by shrinking the coefficients of less important predictors towards zero, allowing for the selection of the most important predictors. Similar to the RFE method, 10 fold cross validation (K fold method) was applied as the resampling method to estimate the performance of the Lasso model. Additionally, RMSE was also calculated to evaluate the model's performance during cross validation. After fitting the Lasso model, predictor importance was

measured based on the absolute value of the t statistic for each predictor. Predictors with larger t statistic values were considered more important. Ultimately, the significant factors that impact the isotopic composition of precipitation at the sampling sites in Southeast Asia were identified by analyzing RFE and Lasso regression models (Table 1).







**Table 1** Optimum predictors selected from RFE technique and/or Lasso regression model.

| Station | Isotope | Method | MJO | PDO | IOD | NAO | QBO | BEST | Wind speed | OLR | Potential evaporation | Vapor pressure | Temperature |
|---|---|---|---|---|---|---|---|---|---|---|---|---|---|
| Bangkok | δ¹⁸O (VSMOW‰) | RFE | | | | | | * | | * | * | | * |
| | | Lasso Regression | | | | | | * | * | * | * | * | * |
| Jakarta | δ¹⁸O (VSMOW‰) | RFE | | | | | | * | | * | | * | * |
| | | Lasso Regression | * | | | | | * | * | * | * | * | * |
| Jayapura | δ¹⁸O (VSMOW‰) | RFE | | | | | | * | * | * | * | | * |
| | | Lasso Regression | * | | * | | | * | | * | * | * | * |
| Kota Bharu | δ¹⁸O (VSMOW‰) | RFE | | | | | * | * | * | | * | * | |
| | | Lasso Regression | * | | | | | * | | * | * | * | |
| Kuala lumpur | δ¹⁸O (VSMOW‰) | RFE | | | | | | * | * | * | | | |
| | | Lasso Regression | * | | * | | | * | * | | * | * | * |
| Singapore | δ¹⁸O (VSMOW‰) | RFE | | | | | | * | | | * | | * |
| | | Lasso Regression | * | * | * | | | * | | * | * | * | * |

| Station | Isotope | Method | MJO | PDO | IOD | NAO | QBO | BEST | Wind speed | OLR | Potential evaporation | Vapor pressure | Temperature |
|---|---|---|---|---|---|---|---|---|---|---|---|---|---|
| Bangkok | δ²H (VSMOW‰) | RFE | | | | | | * | | * | * | | * |
| | | Lasso Regression | * | | | | | * | * | * | * | * | * |
| Jakarta | δ²H (VSMOW‰) | RFE | | | | | * | * | | * | | * | * |
| | | Lasso Regression | | | | | | * | * | * | * | * | * |
| Jayapura | δ²H (VSMOW‰) | RFE | | | | | | * | | * | * | | * |
| | | Lasso Regression | | | | | * | * | * | * | * | * | * |





| | | | | | | | | | | |
|---|---|---|---|---|---|---|---|---|---|---|
| Kota Bharu | δ²H (VSMOW‰) | RFE | | | | * | | | | |
| | | Lasso Regression | * | | * | * | | | * | * | * |
| Kuala lumpur | δ²H (VSMOW‰) | RFE | | | | * | * | * | * | | * |
| | | Lasso Regression | | * | | | | | | | |
| Singapore | δ²H (VSMOW‰) | RFE | | | | * | | * | | * | * |
| | | Lasso Regression | * | * | | * | * | * | * | * | |

**245** **4.2 The importance of predictor variables in influencing target variable/the isotopic composition of precipitation**

Analyzing the relative significance of different predictor variables that impacts the stable isotope contents can present a valuable findings (Fig. 5 and Fig. A1). According to the developed ML models, several factors, including precipitation amount, potential evaporation, vapor pressure, and temperature are the main parameters influencing the isotopic composition of precipitation at most of the studied sites. These factors have been historically identified as significant drivers of the stable

**250** isotope composition in tropical areas (Clark and Fritz, 1997). At tropical stations, the stable isotope composition of precipitation has a fairly strong relationship with air temperature, which is due to the periodicity of monsoon precipitation. However, at non tropical stations, the temperature is one of the main parameters influencing the stable isotope composition of precipitation (Clark and Fritz, 1997).





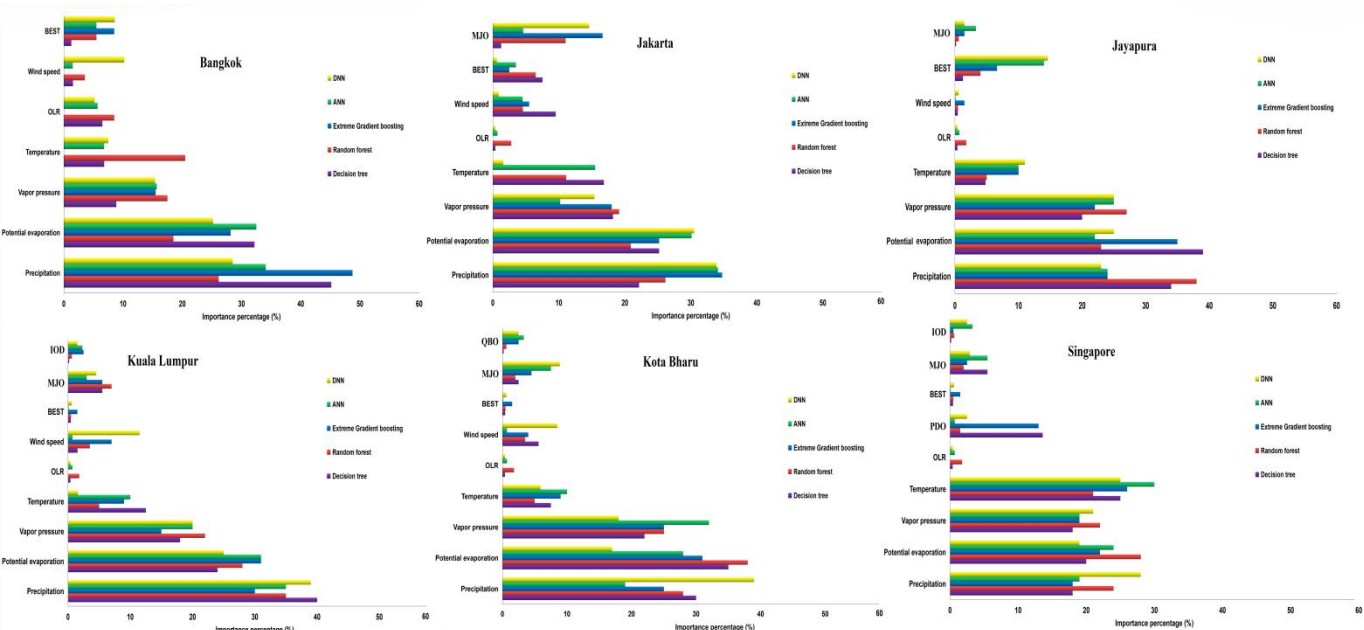

**Figure 5** Fractional importance of various local and regional parameters (predictors) influencing $\delta^{18}O$ content in the studied stations precipitation based on the output from various ML models.

More interesting is the low ranking (much weaker impact) of most regional factors (teleconnection indices) in influencing the stable isotope composition of precipitation. The weak impact of regional factors influencing the stable isotope composition of precipitation compared to the local parameters has also been reported by previous studies in Southeast Asia (Heydarizad et al., 2023b) and other parts of the world (Heydarizad et al., 2021). Previous studies have mentioned the influence of ENSO teleconnection indices on the stable isotope composition of precipitation across Southeast Asia (Heydarizad et al., 2023b; Ichiyanagi and Yamanaka, 2005).

### 4.3 Utilization of various machine learning techniques for predicting stable isotope composition in precipitation

Various machine learning techniques were employed to predict the stable isotope composition of precipitation, while assessing the relative significance of different local and regional factors. The predictors for the ML models were local factors including geopotential height, precipitation amount, potential evaporation, air temperature, vapor pressure, relative humidity, and wind speed, as well as regional factors (teleconnection indices). However, the isotopic composition of precipitation was used as the target variable. The results showed that the models developed based on ML techniques were accurate in most cases due to their high $R^2$ values and low RMSE, NSE, BIC, and AIC values (Table 2). This is due to a much more complicated procedure for processing the data in ML models than regression models. Among the ML models, XGboost showed the highest accuracy in most cases, while DNN demonstrated the highest accuracy in a few cases. The higher





accuracy of the models developed based on XGboost was due to the fact that this model uses a more regularized algorithm that reduces over fitting and gives it much better accuracy. In addition to its higher accuracy, the XGboost model fulfills tasks at a significantly higher speed of up to 10 times faster compared to other ML models, which is due to the fact that XGboost conducts numerous calculations and processes simultaneously (Nishida, 2017).


**Table 2** Evaluating the precision of the ML models using various evaluation metrics.

| Station | Isotope | Method | XGboost | DNN | SNN | Random forest | Decision tree | Isotope | Method | XGboost | DNN | SNN | Random forest | Decision tree |
|---|---|---|---|---|---|---|---|---|---|---|---|---|---|---|
| Bangkok | $\delta^{18}O$ (VSMOW‰) | AIC | 405 | 585 | 607 | 498 | 520 | $\delta^2H$ (VSMOW‰) | AIC | 960 | 989 | 1110 | 1068 | 1140 |
| | | BIC | 410 | 590 | 620 | 512 | 531 | | BIC | 981 | 995 | 1135 | 1072 | 1146 |
| | | $R^2$ | 0.91 | 0.72 | 0.69 | 0.88 | 0.84 | | $R^2$ | 0.87 | 0.80 | 0.55 | 0.64 | 0.33 |
| | | VNS | 0.90 | 0.71 | 0.67 | 0.87 | 0.82 | | VNS | 0.86 | 0.80 | 0.54 | 0.61 | 0.32 |
| | | RMSE | 0.76 | 2.0 | 2.4 | 1.3 | 1.5 | | RMSE | 12.20 | 15.50 | 22.10 | 18.72 | 28.30 |
| Jakarta | $\delta^{18}O$ (VSMOW‰) | AIC | 435 | 545 | 530 | 570 | 690 | $\delta^2H$ (VSMOW‰) | AIC | 973 | 1085 | 993 | 1065 | 1072 |
| | | BIC | 452 | 567 | 542 | 583 | 710 | | BIC | 991 | 1097 | 1012 | 1075 | 1095 |
| | | $R^2$ | 0.89 | 0.75 | 0.76 | 0.73 | 0.32 | | $R^2$ | 0.85 | 0.65 | 0.78 | 0.69 | 0.72 |
| | | VNS | 0.88 | 0.73 | 0.74 | 0.73 | 0.31 | | VNS | 0.85 | 0.64 | 0.77 | 0.68 | 0.70 |
| | | RMSE | 0.91 | 1.6 | 1.6 | 1.8 | 3.3 | | RMSE | 12.80 | 19.20 | 16.20 | 18.10 | 18.60 |
| Jayapura | $\delta^{18}O$ (VSMOW‰) | AIC | 540 | 445 | 521 | 605 | 620 | $\delta^2H$ (VSMOW‰) | AIC | 1090 | 985 | 1040 | 1069 | 1140 |
| | | BIC | 553 | 460 | 536 | 618 | 629 | | BIC | 1110 | 996 | 1062 | 1082 | 1163 |
| | | $R^2$ | 0.75 | 0.87 | 0.76 | 0.68 | 0.65 | | $R^2$ | 0.61 | 0.84 | 0.76 | 0.68 | 0.33 |
| | | VNS | 0.74 | 0.87 | 0.76 | 0.68 | 0.61 | | VNS | 0.60 | 0.84 | 0.74 | 0.65 | 0.31 |
| | | RMSE | 1.70 | 1.10 | 1.5 | 2.6 | 2.7 | | RMSE | 20.10 | 13.15 | 16.90 | 17.80 | 25.5 |
| Kota Bharu | $\delta^{18}O$ (VSMO | AIC | 470 | 535 | 595 | 570 | 624 | $\delta^2H$ (VSMOW‰) | AIC | 985 | 1090 | 1062 | 1083 | 1211 |
| | | BIC | 476 | 543 | 606 | 585 | 635 | | BIC | 996 | 1110 | 1076 | 1097 | 1252 |
| | | $R^2$ | 0.85 | 0.74 | 0.69 | 0.70 | 0.63 | | $R^2$ | 0.84 | 0.61 | 0.63 | 0.62 | 0.32 |





| Station | Variable | Metric | | | | | | | Variable | Metric | | | | | |
|---|---|---|---|---|---|---|---|---|---|---|---|---|---|---|---|
| | W ‰) | VNS | 0.84 | 0.74 | 0.69 | 0.70 | 0.62 | | | VNS | 0.84 | 0.60 | 0.62 | 0.62 | 0.31 |
| | | RMSE | 1.1 | 1.6 | 2.4 | 2.3 | 2.6 | | | RMSE | 13.15 | 20.10 | 18.72 | 19.90 | 27.75 |
| Kuala lumpur | $\delta^{18}O$ (VSMOW ‰) | AIC | 412 | 480 | 509 | 526 | 490 | | $\delta^2H$ (VSMOW‰) | AIC | 942 | 1012 | 1040 | 1085 | 1115 |
| | | BIC | 422 | 486 | 524 | 545 | 502 | | | BIC | 961 | 1026 | 1062 | 1099 | 1176 |
| | | $R^2$ | 0.90 | 0.84 | 0.78 | 0.76 | 0.82 | | | $R^2$ | 0.91 | 0.82 | 0.76 | 0.60 | 0.45 |
| | | VNS | 0.90 | 0.84 | 0.77 | 0.76 | 0.81 | | | VNS | 0.90 | 0.81 | 0.74 | 0.59 | 0.42 |
| | | RMSE | 0.83 | 1.0 | 1.4 | 1.5 | 1.2 | | | RMSE | 10.50 | 14.20 | 16.90 | 20.90 | 24.60 |
| Singapore | $\delta^{18}O$ (VSMOW ‰) | AIC | 446 | 614 | 605 | 533 | 518 | | $\delta^2H$ (VSMOW‰) | AIC | 1024 | 955 | 1052 | 1121 | 1077 |
| | | BIC | 461 | 629 | 619 | 546 | 531 | | | BIC | 1032 | 970 | 1065 | 1135 | 1089 |
| | | $R^2$ | 0.88 | 0.65 | 0.66 | 0.81 | 0.84 | | | $R^2$ | 0.80 | 0.89 | 0.71 | 0.42 | 0.61 |
| | | VNS | 0.87 | 0.64 | 0.66 | 0.81 | 0.83 | | | VNS | 0.78 | 0.88 | 0.70 | 0.41 | 0.60 |
| | | RMSE | 0.93 | 2.91 | 2.90 | 2.2 | 1.9 | | | RMSE | 15.90 | 11.32 | 17.30 | 25.90 | 20.10 |

To ensure the precision of the models, stable isotope contents in precipitation, generated by the most precise machine learning model, have been compared with the measured data at each station in this study. The comparsion results (Fig. 6 and Fig. A2) showed acceptable matching between simulated and measured stable isotope data. While the simulation created by the ML models showed acceptable accuracy, further refinement of these models is also possible. Adding more predictors to the ML models, like cloud microphysical properties including cloud top temperature and cloud top pressure, can improve the accuracy of the models. Nevertheless, these factors only cover a small part of the stable isotope dataset and are not available

for the whole period of the stable isotope data in the studied stations.

Furthermore, the utilization of hybrid algorithms including machine learning-Q statistic algorithms can contribute to developing more precise models.





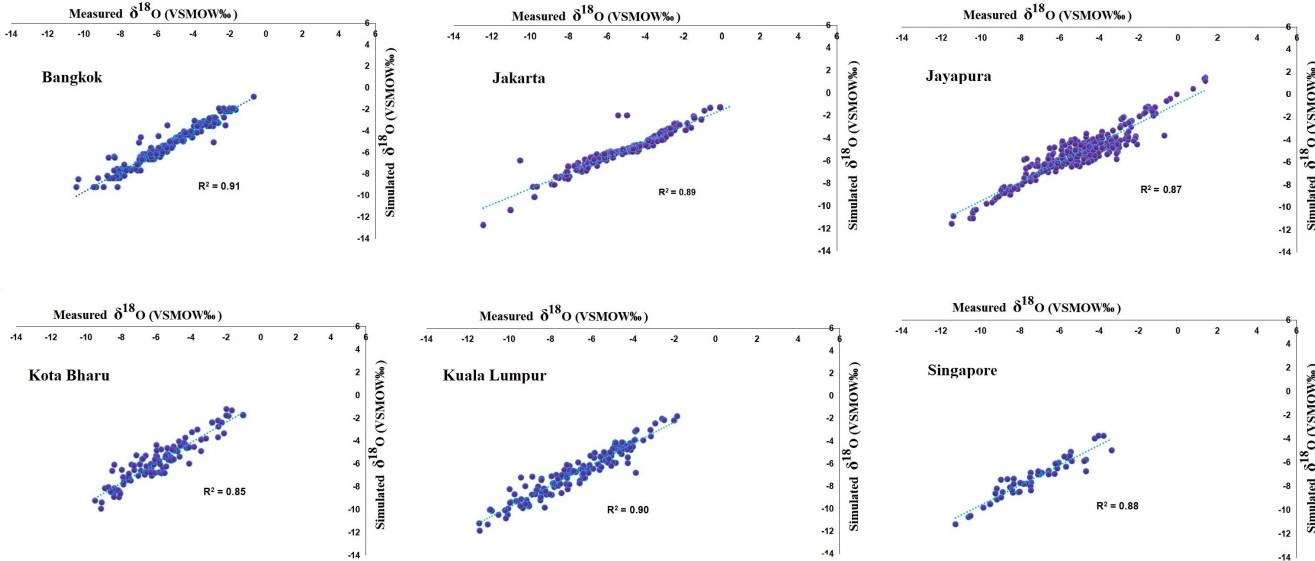

**Figure 6** Examining the differences between measured and simulated δ$^{18}$O content in precipitation using the most accurate ML models by R$^2$ values.

### 4.4 Evaluating model performance in predicting stable isotope contents with Bootstrap confidence intervals

To evaluate the uncertainty in the simulated stable isotope contents of precipitation, a bootstrap technique was utilized. A 95% confidence interval for the predicated data was calculated using this method, which provided a better understanding of the variation of predictions from the developed model to other existing statistics. Figures 7 and figure A3 display the 95% confidence intervals for the stable isotope contents of precipitation at the studied stations. Most stable isotope data fit within the confidence intervals, suggesting that the ML model precisely estimated the stable isotope contents for each station.

However, there were instances where the predicted data surpaseed the upper limit of the confidence interval, showing that the model significantly underestimated the higher values. On the other hand, there were also cases where the data was below the lower boundary of the confidence interval, suggesting that the model had overestimated the very low stable isotope contents.





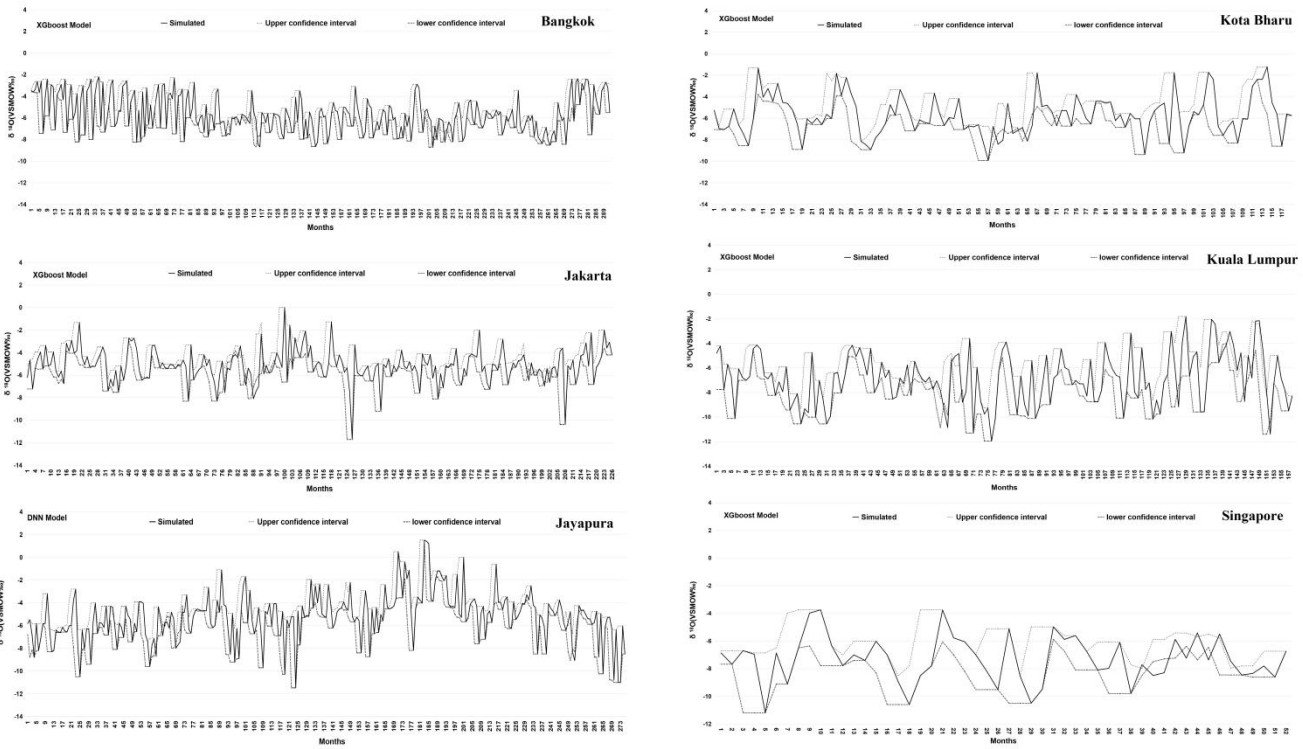


**Figure 7** Examining the differences between measured and simulated $\delta^{18}O$ content in precipitation using the most accurate ML models by $R^2$ values.

### 4.5 Forecasting stable isotope contents in precipitation with VAR and ML models


Finally, the stable isotope composition of precipitation was forecasted for one year using the VAR method and compared with the forecasted stable isotope data using an ML model at the studied stations (Fig. 8 and Fig. A4). The results demonstrated that the ML models could forecast the stable isotope contents of precipitation with higher precision relative to the VAR models in most of the study sites except for Singapore and Kota Bharu for $\delta^2H$ isotope and Jakarta station for $\delta^{18}O$

isotope due to lower RMSE values of ML models compared to VAR model outputs (Fig. A5). This study depicts that ML techniques can forecast stable isotope contents with acceptable accuracy. There are several reasons why ML forecasting is more accurate than other methods. Firstly, ML models can determine patterns that are too complex for other methods to detect. Secondly, ML models usually are more flexible than other techniques and allow the quick infusion of new information into models. Thirdly, unlike traditional methods, ML forecasting algorithms often apply techniques that involve

more complex features and predictive methods compared to other ones which improve the accuracy of forecasts while minimizing a loss function.



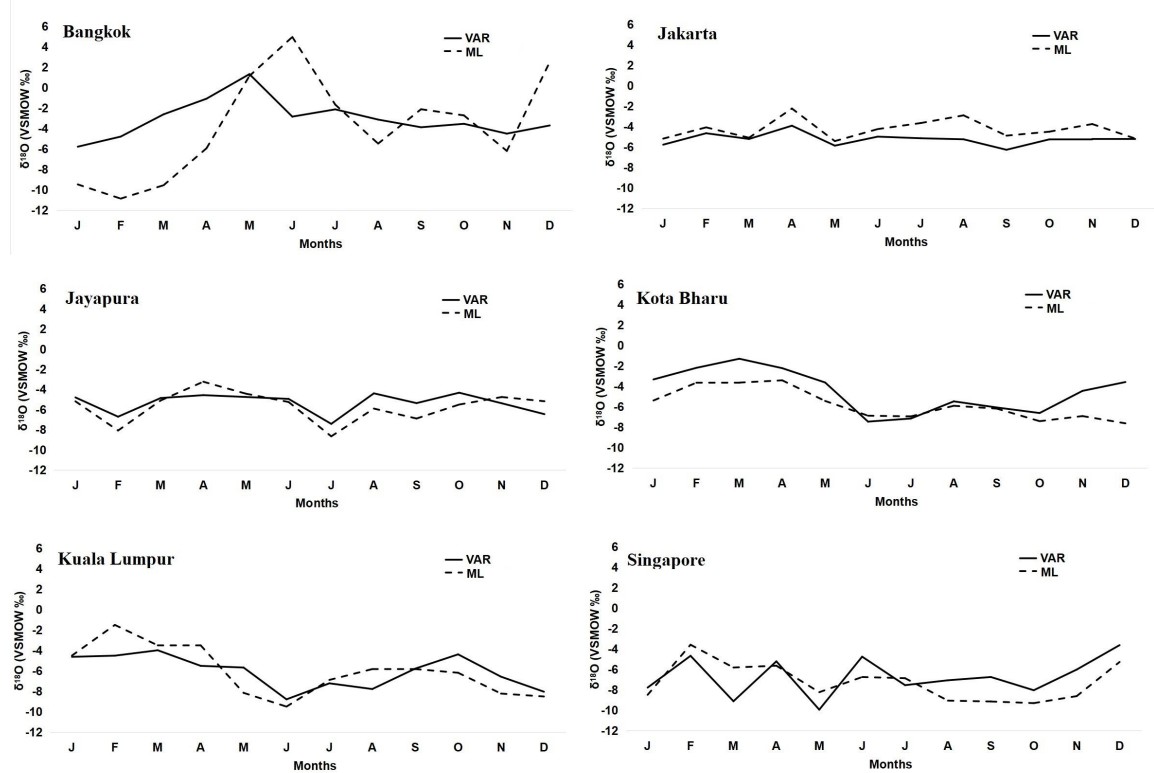

**Figure 8** Comparison of δ¹⁸O content in the studied stations precipitation for 12 months using VAR and ML models.

## 5. Conclusion

The stable isotope composition of precipitation was simulated by diverse ML models at the studied stations in Southeast Asia. The results showed that the XGboost method resulted in more accurate models in most cases according to various evaluation metrics (AIC, BIC, NSE, $R^2$, and RMSE). This study also demonstrated that local and regional predictors influence the stable isotope composition of precipitation of the studied stations. The stable isotope composition of precipitation depends mainly on the vapor pressure, precipitation amount, temperature, and potential evaporation. The results of a bootstrap uncertainty analysis showed that the ML models could predict the stable isotope compositions of precipitation accurately. Finally, the results of stable isotope forecasting using ML and VAR models reveal that ML models are also highly accurate for forecasting stable isotope contents in precipitation compared to the VAR method. This is due to their significant ability to determine patterns that are too complex for other methods to detect as well as their notable flexibility in prediction compared to other techniques.




## Appendix A: Extra figures

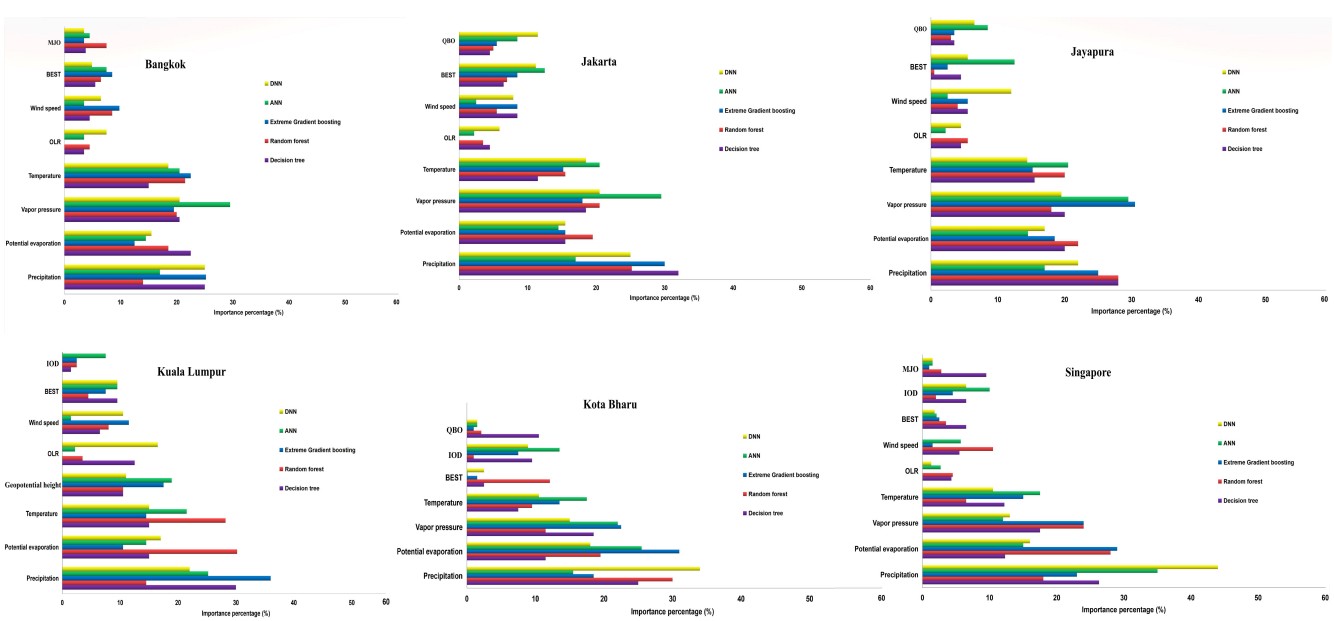

**Figure A1** Fractional importance of various local and regional parameters (predictors) impacting δ²H content in the studied stations precipitation based on the output from various ML models.

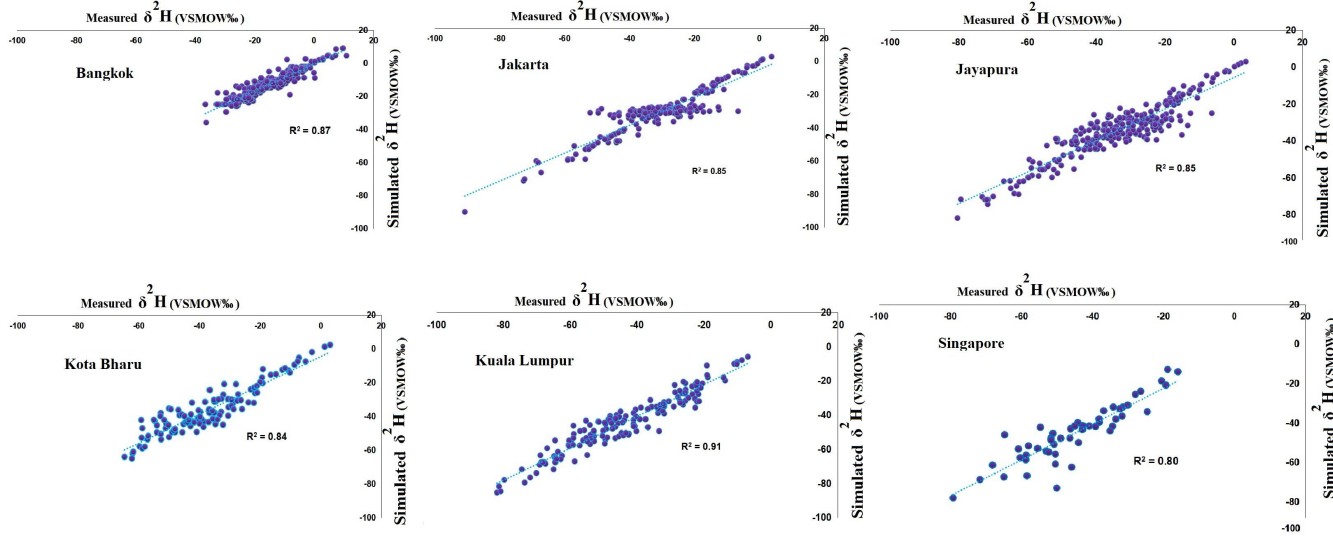

**Figure A2** Examining the differences between measured and simulated δ²H content in precipitation by the most accurate ML models by R² values.






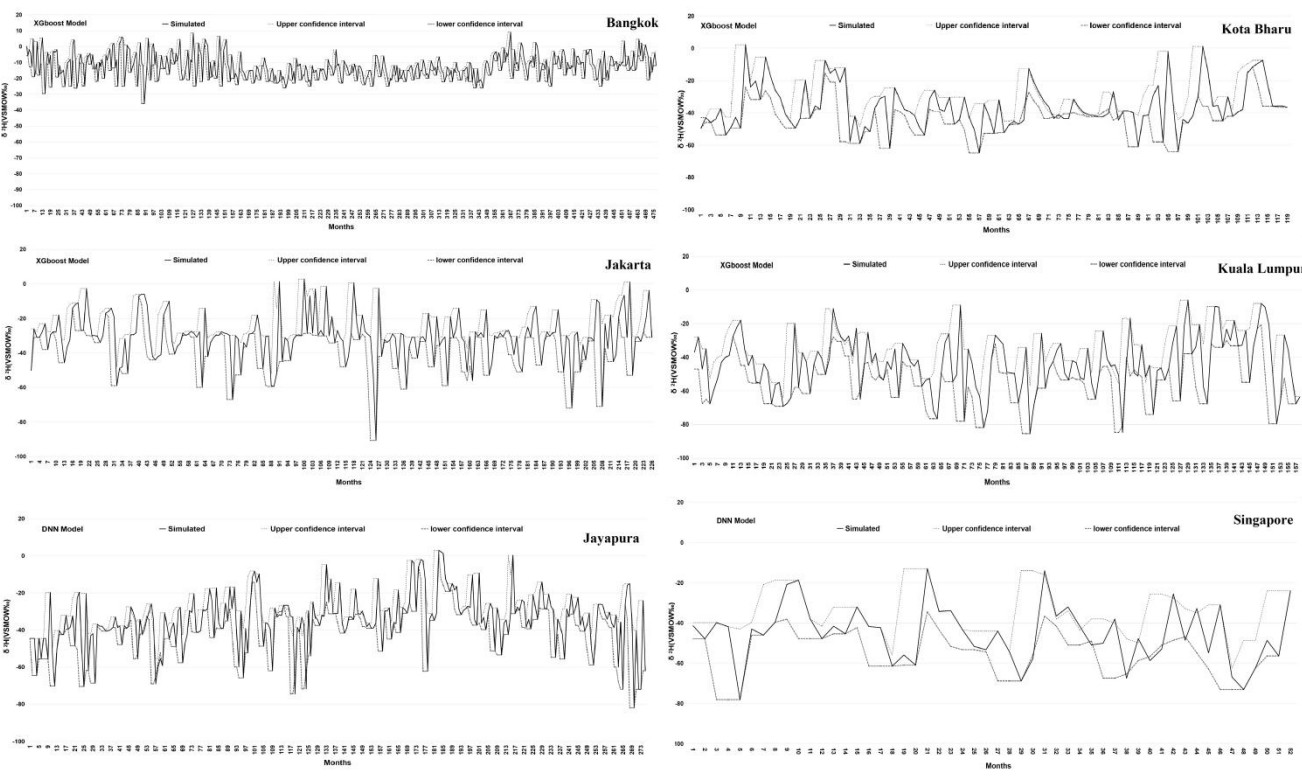

**Figure A3** Confidence intervals by a bootstrap analysis for predicted δ²H content in the studied stations using the most accurate ML model.





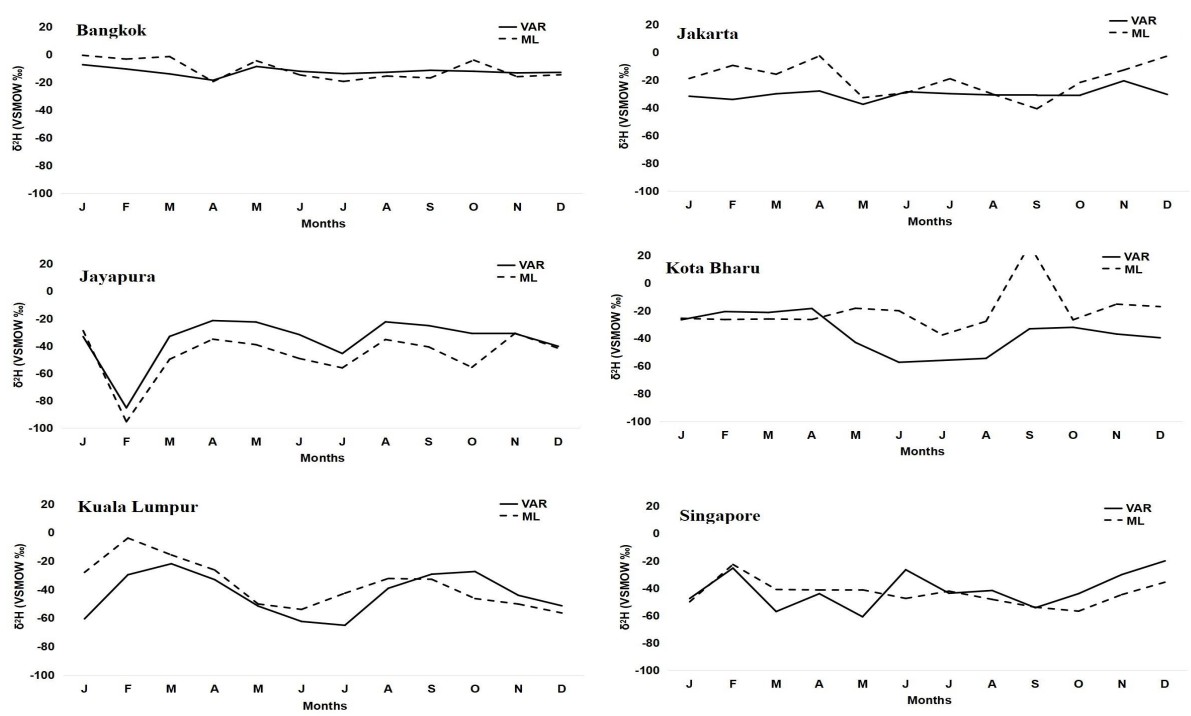


**Figure A4** Comparison of δ²H content in the studied stations precipitation for 12 months using VAR and ML models.

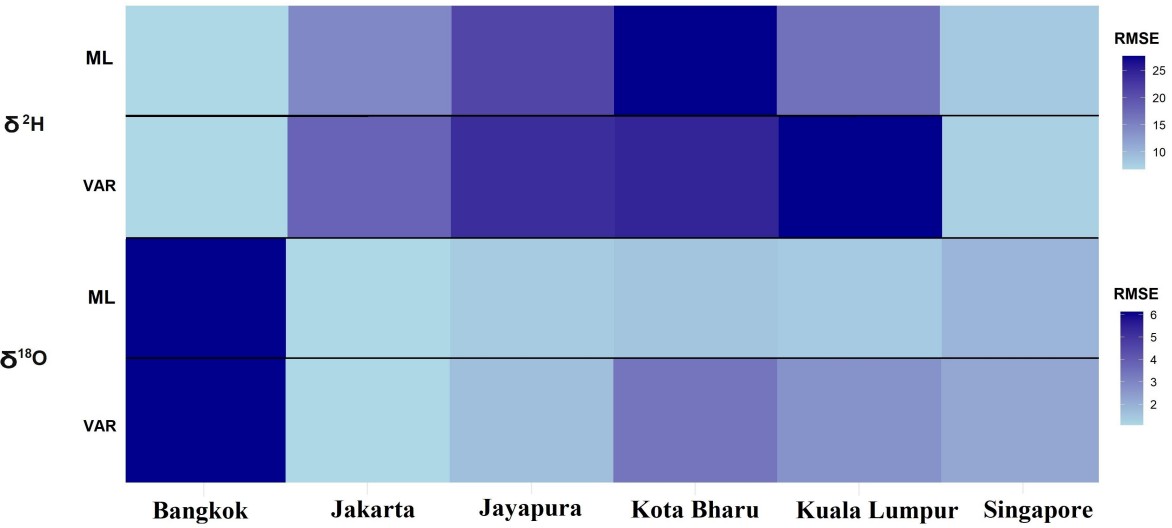

**Figure A5** Performance of evaluated ML and VAR models for the studied stations in Southeast Asia.





**Code and data avilability**

The R software was used to perform all statistical data analyses (version 4.1.3). The R packages used in this study were "devtools", "tidyverse", "corrplot", "caret", "leaps", "MASS", "olsrr", "GGally", "glmnet", "Metrics", "dplyr", "pls", "lattice", "quantreg", "ggplot2", "rsample", "reshape2", "lubridate", "ncdf4", "rts", "ParamHelpers", "data.table", "e1071", "stringr", "readr", "xgboost", "gbm", "h2o", "pdp", "datasets", "caTools", "party", "magrittr", "randomForest", "keras",
"mlbench", "neuralnet", "lime" ,"mc2d", "lhs", "fitdistrplus","boot", "vars", "stats" , "lmtest" , "tseries", "dynlm", and "leaps". The codes used for data processing are available at ……... Data sets used in this study are also available at Global Network of Isotopes in Precipitaion (GNIP) website at  https://www.iaea.org/services/networks/gnip.

**Author contributions**

Conceptualization, Mojtaba Heydarizad and Masoud Minaei; investigation, Mojtaba Heydarizad, Hamid Ghalibaf Mohammadabadi, and Nathsuda Pumijumnong; methodology, Masoud Minaei, Rogert Sori, and Liu Zhongfang; project administration, Nathsuda Pumijumnong and Pouya Salari; software, Pouya Salari and Mojtaba Heydarizad; supervision, Liu Zhongfang; writing—original draft, Mojtaba Heydarizad and Rogert Sori.

**Competing interests**

The authors declare that they have no conflict of interest.

**Acknowledgments**

The postdoctoral fellowship (no. 202323310039) granted by the School of Ocean and Earth Science at Tongji University,
China is acknowledged by the first author, Mojtaba Heydarizad. Rogert Sorí is grateful for the support provided by the postdoctoral contract ''Ramón y Cajal'' no. RYC2021-034044-I, financed by the Ministerio de Ciencia e Innovación of Spain. The authors are grateful to the Global Network of Isotopes in Precipitation (GNIP) for supplying the isotope data of precipitation for the study stations. The authors also would like to express their gratitude to Dr. Elham Mahdipour from the Department of Computer Engineering at Khavaran Institute of Higher Education in Mashhad, Iran, for her invaluable
comments and suggestions during this study.

**Financial support**

This research did not receive any specific grant from funding agencies in the public, commercial, or not for profit sectors.




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
