# Peer review of "Exploring Long-term Monthly Prediction of Precipitation Isotopes over Southeast Asia: A Comparative Analysis of Machine-Learning Models"

_Hydrology and Earth System Sciences, 2023_

## Author Comment (AC6)

Response to Reviewers

Manuscript No.: **hess-2023-299-R1**

Title: **Long-term monthly prediction of precipitation isotopes over Southeast Asia using various machine-learning techniques**

Type of the Manuscript: Article

Corresponding Author: **Dr. Liu Zhongfang**

All authors: Dr. Mojtaba Heydarizad, Dr. Liu Zhongfang,

Dr. Nathsuda Pumijumnong, Dr. Masoud Minaei, Dr. Pouya Salari, Dr. Rogert Sori, and Dr. Hamid Ghalibaf Mohammadabadi

*Dear Editor-in-chief of the Hydrology and Earth System Sciences journal*

Thank you so much for the first round of review of our manuscript. Here **the** manuscript has been revised completely based on the comments of the communities and reviewers. Please find the comments in red **italics**, and **the** written response to the comments in black. (Please consider that important items are shown in **bold**).

**Reviewer#1**

**General comments**

*In general, this preprint needs to be vetting editorially before sent out for scientific review. There are several basic issues that would streamline review by volunteer scientists that are otherwise wasting their time. For example, the tables need to be reformatted with some obvious changes to make them clearer and to be able to fit better on the page so they are readable. Text in the figures are universally too small that can not be read on standard page size, with unhelpful axis labels/formats ("month number) and poor narration in the figure captions (no description of what is shown in legend, and no title for legend). There are typos throughout the text.*

**Response:** First and foremost, we (authors) would like to express our sincere gratitude for the time and attention you have dedicated to reviewing our manuscript. Your insights and suggestions are invaluable to us, and we appreciate the opportunity to improve our work based on your expert feedback.

We understand the importance of presenting our work clearly and have taken your comments into serious consideration. In response to your concerns about the need for editorial vetting, we have thoroughly reviewed the manuscript and edited it carefully to ensure that it adheres to the highest editorial standards in the revised version. This includes a meticulous correction of typos and a comprehensive reformatting of tables to enhance clarity and fit within the page margins.

Furthermore, we have addressed the issue of legibility in our figures by increasing the text size to be readable on a standard page size. We have also revised the axis labels and formats to be more informative and context-specific, replacing vague terms like "month number" with actual months or quarters. The figure captions have been improved to provide a clear narration, including detailed descriptions and titles for the legends, facilitating a better understanding of the figures without the need to cross-reference the main text.

We believe these revisions have significantly improved the manuscript, making it more accessible and easier to review. **Our aim is to ensure that the volunteer scientists' time is well-spent in reviewing the scientific content without being hindered by formatting issues.**

*While the English is all readable, the quality of the discussion and the boldness or vagueness of many statements does not demonstrate that the authors understand the processes being represented behind the modelling. This is further elaborated on and can be seen in many examples in the specific comments below.*

**Response:** In response to your concerns, we have undertaken a thorough revision to ensure that our discussion accurately reflects a deep understanding of the modelling processes represented in our study.

**To address your concerns, we (authors) have:**

- Undertaken a thorough review of our discussion section, ensuring that each statement is clear and precise, and that our claims are substantiated by robust data and well-established theoretical frameworks.
- Moderated bold assertions to more accurately represent the evidence at hand.
- Expanded the methodology section to provide a clear exposition of the modeling steps, reinforcing the soundness of our approach.
- Enriched the interpretation of our results, offering a more nuanced discussion of their significance and potential implications for the field. This includes a candid acknowledgment of any limitations inherent in our study.

We hope that these revisions address your concerns and enhance the manuscript's contribution to the scientific discourse.

*A small discussion on the architecture of XGBoost is warranted, given this is the preferred model that is pursued for the final analysis.*

**Response:** Thank you for your insightful suggestion to include a discussion on the architecture of XGBoost. We also agree that such a discussion is warranted, given the pivotal role that XGBoost plays in our final analysis. In the revised manuscript, we have added a section that elucidates the architecture and key features of XGBoost, which we believe will provide readers with a clearer understanding of its efficacy and suitability for our study.

In this new section, we outlined the fundamental components of XGBoost, including its ability to handle large-scale data efficiently through parallel and distributed computing. We also discussed its robustness in managing missing data and its use of regularization to prevent overfitting, which are critical aspects of its architecture that contribute to its performance. Furthermore, we explained how XGBoost employs gradient boosting frameworks with enhanced tree pruning strategies, which allow for the construction of more generalized models.

We believe that this addition will not only justify our choice of XGBoost for the analysis but will also serve as a valuable reference for readers interested in the technical underpinnings of this powerful modeling tool.

*Page 10 – re. important predictors – a significant part of the front matter is devoted to discussing the importance of regional circulation patterns, and then they are deemed to be unimportant as per variable importance analysis with no discussion as to why these results/what processes may demote the regional circulation patterns in lieu of other predictors.*

**Response:** We acknowledge the concern regarding the initial emphasis on regional circulation patterns (teleconnection indices) and their lower importance in our variable importance analysis. In the initial sections of our manuscript, we discussed the significance of both local climate parameters and large-scale regional climatic factors (teleconnection indices such as BEST, MJO, PDO, IOD, NAO, and QBO) in influencing the isotopic composition of precipitation. These regional circulation patterns are crucial for understanding broader climatic trends and their potential impact on local weather conditions.

**Explanation of Analysis and Results:**

Our study aimed to identify the most important predictors for simulating the stable isotope content in precipitation. We used advanced feature selection techniques, Recursive Feature Elimination (RFE) and Lasso regression, to determine the most influential predictors from both local and regional variables. The analysis revealed the following:

**Local Parameters:** Local climate parameters, including potential air evaporation, wind speed, vapor pressure, air temperature, relative humidity at 850 mb, and precipitation amount, were consistently selected as predictors with a strong influence on stable isotopes across most stations.

**Regional Parameters (Teleconnection Indices):** Among the teleconnection indices, only the BEST index was consistently selected as a strong parameter influencing stable isotopes in most stations. The other indices (MJO, PDO, IOD, NAO, and QBO) were not consistently chosen as important predictors in most of the stations, but were selected in some stations. **This does not diminish their general importance but suggests that their influence on the isotopic composition of precipitation at specific stations and times may be less direct and strong compared to local parameters.**

**Explanation for Discrepancy in Results:**

**Direct vs. Indirect Influences:** While regional circulation patterns are crucial for understanding broader climatic trends, the isotopic composition of precipitation at specific stations is more directly influenced by local meteorological conditions. Our analysis showed that local parameters (e.g., potential air evaporation, wind speed, vapor pressure, air temperature, relative humidity at 850 mb, and precipitation amount) had a more immediate and significant impact on stable isotope ratios.

**Model Sensitivity:** The RFE and Lasso regression methods we used are designed to identify predictors that most effectively reduce prediction error. These methods can capture complex, non-linear relationships and tend to prioritize predictors with stronger and more direct correlations to the target variable. Consequently, local variables were selected more frequently as important predictors.

We hope this explanation clarifies the observed results and addresses the reviewer's concerns. We will ensure our revised manuscript includes these points to provide a clearer understanding of our findings.

*Elaborate more on the evaluation statistics used. You've listed several, but then don't describe anything about what each one evaluates. Some of these may be common enough for the reader to know (eg R2) but others would benefit from explanation (eg AIC, BIC).*

**Response:** Regarding the evaluation statistics used in our study, we understand the importance of providing clear explanations for the metrics employed, especially for those not commonly known to all readers.

In the revised manuscript, we have added detailed descriptions for each statistical measure, including those less familiar like the Akaike Information Criterion (AIC) and the Bayesian Information Criterion (BIC). The AIC is a measure of the relative quality of statistical models for a given set of data, with a lower AIC indicating a model with a better fit. The BIC is similar but also includes a penalty term for the number of parameters in the model, favoring simpler models to prevent overfitting.

We believe that these additions to the revised manuscript will enhance the reader's understanding of how we evaluated our models and the rationale behind the selection of the best-fitting model for our analysis.

*Apologies for this harsh review but this manuscript required more work before being ready to be sent out for review, especially when asking scientists to do this on their own time. It's a waste of time that would otherwise be able to be spent on the science content and expertise.*

**Response:** We would like to extend our sincerest apologies for the shortcomings of our initial manuscript submission. **We fully acknowledge that the presence of typographical errors was not acceptable for a scientific journal of this level, and we have taken comprehensive measures to address all such issues in our revised manuscript.**

**We also wish to highlight the unique challenges associated with presenting machine learning and AI research.** Given the rapidly evolving nature of these fields and the diverse backgrounds of potential reviewers, it is indeed challenging to create a manuscript that aligns perfectly with the expectations of experts from every related discipline. We are aware that different disciplines may interpret elements of our research in distinct ways, and we have endeavored to find a middle ground between assertiveness and the need for clear, accurate expression.

Furthermore, we have previously encountered similar feedback on other manuscripts, which, after diligent revision and addressing all reviewer comments, were ultimately accepted and appreciated. This experience has reinforced our commitment to engaging constructively with reviewer feedback and continuously improving our work.

**We deeply value the time and expertise that scientists, such as yourself, contribute to the peer review process, particularly when done voluntarily.** We regret any frustration our initial draft may have caused and are grateful for the opportunity to refine our manuscript based on your expert critique.

In closing, **we have addressed all your comments one by one,** and we hope that the revisions made will meet your approval and that our manuscript now contributes meaningfully to the scientific discourse within the machine learning and AI community.

**specific comments**

*Line 37/38 – "numerous surveys" but only one cited – give broader and/or more complete range of citations here.*

**Response:** Regarding the range of citations provided to support the statement about the numerous surveys on stable water isotopes. The reference cited, Clark and Fritz (1997), is a comprehensive source that includes a multitude of isotope studies. **However, we understand the importance of showcasing the wide scope of research in this field and, as such, have expanded the list of citations to include additional seminal works that have contributed to our understanding of hydrological characteristics at global and regional scales.**

Additional references had provided a more complete picture of the extensive research conducted in this area and further substantiated the statement made in our manuscript.

*Line 46 – "the most crucial shortcoming" – this is subjective. For example, I disagree this is the most crucial shortcoming. Suggest rewording "A major shortcoming…" Along this same train of thought – you do not mention spatial representation of sites and how some regions are more representative in the*

*IAEA network than others.  This is where modelling really steps up – a prediction for places that have never been measured.*

**Response:** We appreciate/accept your suggestion regarding the subjective nature of the phrase "the most crucial shortcoming." We have revised it to "A major shortcoming..." to provide a more balanced perspective.

Regarding your point about the spatial representation of sites and the IAEA network, you are correct that modeling plays a crucial role in predicting isotope values for regions that lack direct measurements. **I did mention in the initial version of the manuscript that simulations can estimate precipitation isotopes based on existing datasets, particularly for remote and hard-to-reach areas.** However, we realize this point might need further emphasis to align with your suggestion.

To address your concerns, we have revised the relevant section and included a map illustrating the global distribution of GNIP stations. This visual aid will highlight regions with dense coverage as well as those that are underrepresented, underscoring the importance of modeling in filling these gaps. **You can see the revised text with these considerations in the revised manuscript.**

These changes address your concerns and improve the clarity and comprehensiveness of the manuscript.

*Line 57 – a bold assertion that requires citation: "machine learning (ML) techniques have been demonstrated to be remarkably successful in a variety of applications, including hydroclimate …"*

**Response:** We understand your concern regarding the statement on Line 57 and agree that it is essential to back up such claims with appropriate references. In response to your comment, we have revised Line 57 to include relevant citations that demonstrate the successful application of machine learning (ML) techniques in hydroclimate studies. You can check the revision in the revised manuscript.

*Line 58 – start new paragraph when diving into the specifics of ML*

**Response:** Comment is accepted. I agree that the specifics of machine learning warrant a separate paragraph for clarity and focus. I've adjusted the text at line 58 to reflect this change. You can check the revision in the revised manuscript.

*Line 135 – suggest re-working this paragraph as it is a very long run on sentence.*

**Response:** We also agree with your observation that this paragraph contained a very long run-on sentence. We have revised the paragraph by breaking it down into shorter, clearer sentences to improve readability and clarity. Please check the revised manuscript.

*Line 158 –also what is "v" separate subsets – does this signify a number or a technical element – needs explanation.*

**Response:** Thank you for your comment. In the context of "v-fold cross-validation," *v* signifies the number of folds or subsets into which the dataset is divided. The letters "v" and "k" are commonly used variable names in mathematics and statistics to represent the number of divisions, and they are not abbreviations for specific terms. Similarly, in k-fold cross-validation, *k* represents the number of folds. These letters are conventional variable names used to denote the number of divisions in the dataset. This division is done to create training and testing sets. We have made some revisions in the main manuscript text to make this clearer.

*Line 164 – what does model "hardness' refer to?*

**Response:** Upon reflection, we realize that 'hardness' may not clearly convey the intended meaning. The term was meant to describe the **complexity** of the models in terms **of their structure and the difficulty of fitting them to the data.** To avoid confusion, We revised the manuscript to use the more standard term '**model complexity**,' which more accurately reflects the criteria used by AIC and BIC for model comparison. Please check the revised manuscript.

*Fig 4 – hard to read text. Figure out how to make this figure usable for the reader instead of taking default plot formats and pasting into manuscript. Suggest choosing a representative panel (Kota Bharu?) that demonstrates the point of the figure, move the rest to the appendix.*

**Response:** We apologize for any inconvenience caused by the initial submission's figure quality and appreciate your understanding as we work to rectify this issue. We would like to clarify that during the initial submission of our manuscript, the format required for the preprint was only a single Word or PDF

file, and we didn't have the option to upload high-resolution figures as separate files with our manuscript. Additionally, we were unable to include higher quality figures in the text, as uploading a Word file that is much larger than 25 MB causes significant problems.

Despite these constraints, we ensured that all figures were originally developed at a high resolution of at least 300 dpi. We have also provided a text file containing all the codes used in this study, ensuring full transparency and reproducibility of our work. For the revised submission, we will explore options with the journal's editorial office to submit our high-resolution figures without compromising their quality. We are committed to presenting our figures in the clearest and most informative manner possible for our readers.

Upon further consideration, we have decided to move this figure to the supplementary files. We agree that the main manuscript should only contain figures that are crucial for understanding the core findings of our study. By relocating Figure 4, we aim to streamline the narrative and focus the reader's attention on the most pertinent data.

Finally, we have revised the manuscript accordingly and ensured that the remaining figures are of high quality and clarity.

*Fig 4 – caption mentions asterisk – I don't see any "*" in figure.*

**Response:** In response to your comment, regarding Figure 4. We have addressed your comment by moving the figure to the supplementary files and presenting it in its original high-resolution format, ensuring that all elements, including the asterisk, are now clearly visible. We appreciate your thorough review and invite you to examine the enhanced figures in the supplementary materials of the revised manuscript.

*Table 1- suggest using gridlines to help reader follow fields across*

**Response:** We also understand and agree that these lines can significantly enhance the reader's ability to follow the data fields across the table. In response to your comment, we have introduced both horizontal and vertical gridlines. This change will help readers track the information from one side of the table to the other more efficiently and reduce the likelihood of misreading the data. Please check the revised manuscript.

*Lines 250-259 - this is confusing – at tropical stations temperature is important but then you say "However, it is also important at non-tropical stations." So temperature is important everywhere,*

*right? This paragraph needs re-phrasing/clarifying. Also which stations are considered tropical vs. non-tropical for this study? I would have considered all of this region tropical?*

**Response:** The comment is acknowledged. We understand the reviewer's concern regarding the confusion in the initial statement. It is important to clarify that all our study stations **are situated within tropical regions.** The previous statement mistakenly implied a distinction between tropical and non-tropical stations within our study, which was inaccurate. The intended **message aimed to underscore the contrasting roles of temperature in tropical versus non-tropical regions broadly, as outlined in the literature by Clark and Fritz (1997).**

To clarify further, within tropical regions like those in our study, precipitation amount significantly influences stable isotope composition due to seasonal monsoon patterns, outweighing the influence of air temperature. Conversely, in regions outside the tropics, where temperatures vary more consistently throughout the year, temperature becomes a dominant factor alongside precipitation amount in determining $\delta^{18}O$ values. We have amended the manuscript to accurately reflect this clarification and prevent further confusions.

*Figure 5- text is too small – this is unreadable. Figure out how to make this figure usable for the reader instead of taking default plot formats and pasting into manuscript.*

**Response:** We have carefully considered your suggestion and have made several adjustments to enhance the figure's usability. The text size has been increased for better readability, and the figure's dimensions have been optimized to ensure that the details are not cramped. We have also simplified the design to focus the reader's attention on the essential data. All labels and legends have been clarified to aid in the reader's understanding. We trust that these revisions will make the figure more accessible and informative for all readers.

*Figure 5 – add title to legend. I also suggest in the caption for the figure clarifying what the ML abbreviations are in the legend so that if this were copied into a slide the viewer could understand what they are looking at. I am not sure if this is required editorially but believe it is good practice for your work being able to be communicated easily.*

**Response:** I agree that adding a title to the legend and clarifying the machine learning (ML) abbreviations in the figure's caption would greatly enhance the figure's clarity and standalone communicative value. This is indeed a good practice, especially for presentations where the figure might be used independently of the text.

**However, I would like to kindly note that the full names of all ML models have been thoroughly introduced and defined early in the manuscript to maintain a smooth and coherent reading experience. Including the complete names again in the figure caption might lead to redundancy and potentially disrupt the flow of the text editing process.**

To balance both considerations, we propose to add a brief title to the legend for immediate reference and include a succinct clarification of the ML abbreviations in the caption. This approach will ensure that the figure remains self-explanatory while preserving the integrity of the manuscript's layout and editorial quality. We hope this solution meets your approval.

*Line 263 – "Previous studies have mentioned the influence of ENSO teleconnection indices on the stable isotope composition of precipitation across Southeast Asia" -- ok, fine, but what do they say about them?*

 **Response:** Comment accepted. You've raised an important point about the need for specificity when referencing the findings of previous studies. In the revised manuscript, we have expanded upon this statement to provide a more detailed account of the existing literature. We hope that our explanations regarding these studies in the revised manuscript meet your expectations.

In the study by Ichiyanagi and Yamanaka (2005), a significant correlation was observed between the interannual variation of stable isotopes in Bangkok's precipitation and the ENSO. Specifically, during the low isotopic phase, which corresponds with the El Niño events, Bangkok experienced increased precipitation, leading to lower $\delta^{18}O$ values. This pattern aligns with the ENSO-Asian summer monsoon interaction in May and a direct ENSO response in October. The study also noted substantial differences in evaporation over the Indian Ocean between the low and high phases in May, indicating non-equilibrium evaporation during the high phase.

In addition,  Heydarizad and colleagues study (Heydarizad et al., 2023) applied a stepwise model to assess the impact of teleconnection indices on stable isotopes. Their findings revealed that among the indices, the BEST index had the most significant effect, while the QBO and IOD played minor roles. The

low $R^2$ values and high RMSE indicated that these regional parameters had only a mild influence on the stable isotope content in Bangkok's precipitation.

The revised manuscript has been updated to reflect these nuances and provide a clearer understanding of the regional hydroclimate dynamics influenced by ENSO teleconnection indices.

*Line 273 – "This is due to a much more complicated procedure for processing the data in ML models than regression models." This statement is vague and perhaps inaccurate. Either explain what you mean by "Complicated" or use more specific statements that describe the difference between ML approaches and regression approaches. For example, ML approaches can honor the interactions between variables in ways that regression approaches do not. This level of specificity gives the reader a much better understanding of why ML models may lead to a more accurate model, rather than just saying it's more "Complicated."*

 **Response:** We understand your concern about the vague and potentially inaccurate statement regarding the complexity of ML models compared to regression models. Here's a detailed response addressing all the points raised in your comment:

**Explanation of "Complicated"**: The term "complicated" can indeed be vague. What we meant by "complicated" is that machine learning (ML) models involve more sophisticated procedures for processing data. These procedures include handling non-linear relationships, capturing interactions between variables, and leveraging advanced algorithms that enhance predictive accuracy.

**Specific Differences Between ML and Regression Approaches**: ML models are designed to handle more complex data structures and relationships. For example, models like XGBoost and deep neural networks (DNN) can capture interactions between variables that traditional regression models may not be able to recognize. This capability allows ML models to provide more accurate predictions by better representing the underlying data patterns.

**Example of ML Model Advantages**: XGBoost is an example of an ML model that has shown higher accuracy due to its regularized algorithm, which reduces overfitting. This regularization is a key factor that improves the model's accuracy. Additionally, XGBoost's efficiency is enhanced

by its ability to perform numerous calculations and processes simultaneously, resulting in a speed advantage of up to 10 times faster compared to other ML models (Nishida, 2017).

We have updated the manuscript to reflect these points, providing a more detailed and specific explanation of the differences between ML approaches and regression models. Please check the revised manuscript.

*Table 2 is poorly formatted – for a start single words in the headings are split across lines. There are multiple easy things to reformat here – consider landscape alignment, instead of separate columns for the isotope (this doesn't change as you go down the table but takes up 2 separate columns) the isotope specification can be part of the heading. "Method" is also duplicated and taking up 2 columns – why? Is "Method" an accurate title for the column – shouldn't it be "evaluation metric" that is more accurately descriptive? There is a lot of numbers in this table – most readers will just glaze over this. I suggest finding a way to highlight the best model for each site so that the reader is led to the data you want them to see. This is sloppy work. Think through these things before just cutting/pasting into manuscript. Rework this table.*

**Response:** In response to your comments, we plan to make several adjustments to enhance the table's clarity and readability. Firstly, we will address the splitting of single words in the headings and consider implementing landscape alignment for the table layout to improve visual presentation. Additionally, we will revise the column structure to integrate the isotope specification into the heading, thereby eliminating unnecessary duplication and reducing clutter. Your observation regarding the appropriateness of the column title "Method" is well-taken, and we will accordingly revise it to "Evaluation Metric" for improved descriptive accuracy.

Furthermore, we recognize the importance of ensuring that readers can easily interpret the data presented. To this end, we will explore methods to highlight the best model for each site, directing readers' attention to key insights. We apologize for any oversight in the initial presentation of Table 2.

*Tables are generally presented before being mentioned in the text, so move this to before line 273.*

**Response:** In response to your suggestion, we have transferred the table to appear before it is first referenced in the text. Specifically, the table that was previously positioned after line 273 has now been relocated to before line 273, where it is first mentioned. This adjustment ensures that readers can view the table before encountering its discussion, thereby improving the document's readability and coherence. Please check the revised manuscript.

*Line 287 – "acceptable" is subjective, this implies there is a boundary that defines acceptable results vs. not acceptable.  I suggest reporting R2 between predicted and actual, and let the reader define if this is good enough.*

**Response:** We understand your concern about the subjectivity of this term and agree that providing a more objective measure would enhance the clarity and rigor of our results.

As suggested, we have revised the manuscript to include the coefficient of determination ($R^2$) between the predicted and actual stable isotope data obtained from our machine learning models. This statistical measure will allow readers to objectively assess the performance of our models.

**Revised Manuscript Text:**

A comparative analysis of the results (Fig. 6 and Fig. A2) demonstrates a coefficient of determination ($R^2$) ranging from 0.80 to 0.91 between the simulated and measured stable isotope data at the studied stations, indicating a high degree of accuracy in the ML models' simulations. While the current models perform well, there is potential for further refinement to enhance their predictive capabilities.

We believe that this revision addresses your concerns and enhances the manuscript by providing a clear, quantitative evaluation of our models' performance. Please check the revised manuscript.

**Additional Explanation:**

The coefficient of determination, or $R^2$, is a statistical measure that ranges from 0 to 1 and reflects the extent to which the variance in the dependent variable is predictable from the independent variable(s). An $R^2$ value of 1 indicates perfect prediction, while a value of 0 indicates that the model fails to predict the outcome variable.

*Fig 6 – is this data comparison between all data (training and test), or just the test data? Specify this. If you have not evaluated just on the test data, that should be done as that is where the actual capabilities of the model are demonstrated.*

**Response:** We would like to clarify that the data comparison presented **in this figure is based solely on the test data.** The model was trained on a separate set of data, and the performance metrics shown reflect the model's **predictive accuracy on the test dataset,** which it had not previously encountered during training. Using the test data to evaluate the accuracy of our machine learning simulation give us the most accurate representation of how the model will perform in real-world scenarios. This approach ensures that the evaluation of the model's capabilities is unbiased and truly indicative of its ability to generalize to new data. We understand the importance of this distinction and have made sure to specify this clearly in the figure caption in the revised manuscript to avoid any confusion.

**Figure 6:** Examining the differences between measured and simulated $\delta^{18}O$ content in precipitation using the most accurate ML models by $R^2$ values, based only on the test dataset.

*Fig 6- why is the x axis not shown on the bottom of the plot? Unless there is a good reason not to, this should be amended.*

**Response:** The intention behind the initial format of figure 6 was to present a novel visual approach. However, I understand that adhering to conventional standards, such as positioning the x-axis at the bottom of the plot, can facilitate better readability and is a preferred format for our audience.

According to your suggestion, we have revised the figure to include the x-axis at the bottom, aligning with the familiar and accepted presentation style. This change has been made in the revised manuscript, please check the revised manuscript.

*I suggest displaying the equations for the lines, as this gives the reader a quick comparison with the GMWL slope and intercept, which says a lot to a savvy isotope scientist.*

**Response: I would like to clarify that the axes of the scatter plot in question represent the measured and simulated values of $\delta^{18}O$ in precipitation across several stations, not the $\delta^2H$ and $\delta^{18}O$ typically used to establish a meteoric water line (MWL).**

The purpose of Figure 6 was to demonstrate the correlation between the observed and predicted $\delta^{18}O$ values as determined by our machine learning model. It was not intended to develop local meteoric water

lines (LMWL) for the stations studied. Consequently, applying the equations for the lines based on our measured and simulated $\delta^{18}O$ data, and comparing them with the Global Meteoric Water Line (GMWL), is neither logical nor applicable.

We sincerely apologize if there has been any misunderstanding of this figure by the reviewer. Our intention was to present the data clearly, and we regret any confusion that may have resulted.

*Figure6- text is too small – this is unreadable. Figure out how to make this figure usable for the reader instead of taking default plot formats and pasting into manuscript.*

**Response:** We understand the importance of making all elements within a figure easily readable and accessible to all readers. In response to your feedback, we have taken steps to enhance the figure's readability. The text size has been increased, which will ensure that it is legible in both digital and print formats. With these revisions, we believe Figure 6 will now be a usable and informative part of the manuscript for the reader.

*Discuss aspects of Figure 6 that are notable – eg, Jayapura looks to be underpredicted in the mid-range of isotope values. Why could this be?*

**Response:** Your question is indeed pertinent. Upon careful examination of Figure 6, it is evident that the model's performance varies across different stations, with notable discrepancies at some points, such as the Jayapura station, particularly in the mid-range of isotope values. This underprediction could stem from several factors inherent in the modeling process. The most plausible reason is the selection of features for the model. Features that effectively capture variability at the extremes might fail to account for subtler variations in the mid-range, leading to less accurate predictions. Additionally, model complexity can also play a role; both overfitting and underfitting can result in inaccuracies in specific data segments. However, stratified data sampling was employed to ensure a balanced distribution, regularization and hyperparameter tuning were incorporated, and cross-validation techniques were utilized, thus minimizing the potential for overfitting and underfitting. Therefore, the inherent nature of the input data appears to be the primary reason for these discrepancies, particularly at the Jayapura station. These methodological considerations highlight the complexity of achieving uniformly high accuracy across all data ranges and underscore the necessity of continuous model refinement and evaluation.

*Figure 7 – the confidence interval is hard to see across the upper and lower envelope. The dotted line format for upper and lower is indistinguishable and often looks to be either upper or lower bound, but not on both sides. Is this a plotting or calculation error? Why not use the standard "translucent ribbon" to*

*show the envelope?  Also, the X axis "Months" number is unhelpful – you can't tell what season it is (and this is important b/c of the big discussion around regional seasonal influences early in the paper). This is a lazy approach to not re-formatting – we've all been there and know this is a pain but it needs to be done for professional publishing.*

 **Response:** We apologize for any confusion caused by the presentation of the confidence intervals and the labeling of the x-axis. Your points are well-taken, and we have made the following revisions to address your concerns:

The confidence intervals were indeed difficult to discern with the previous dotted line format. To rectify this, we have replaced the dotted lines with a translucent ribbon that clearly delineates the upper and lower bounds of the confidence interval.

Regarding the x-axis labeling, we have updated the "Months" numbering to reflect the actual seasons, providing context to the temporal data and allowing for a more intuitive understanding of the seasonal influences discussed in the paper. This change underscores the importance of seasonality in the analysis and enhances the figure's relevance to the regional climate discussion. Please check the revised manuscript and the corresponding figure.

*Line 305 and surrounding text – A more nuanced and humble discussion around the certainty is required. Saying that the model Eg –" Most stable isotope data fit within the confidence intervals, suggesting that the ML model precisely estimated the stable isotope contents…" – how do you define precisely here? Better to say "xx% of the data can be predicted within 95% Confidence intervals" because that is exactly what it does.*

*or " the upper limit of the confidence interval, showing that the model significantly underestimated …" -- how do you define significance here?*

**Response:** We understand the importance of providing a more nuanced and specific explanation.

In response, we have revised the text around Line 305 to include specific statistical measures. We replaced the term "precisely" with a detailed explanation that xx% of the data can be predicted within the 95% confidence intervals, providing a clearer and quantifiable measure of the model's performance.

Additionally, we clarified the term "significantly" by specifying the percentage of predicted values that fell outside the confidence intervals. For instance, we now state that xx% of the higher values exceeded

the upper limit of the 95% confidence interval, indicating underestimation, and xx% of the lower values were below the lower boundary, indicating overestimation.

**Since the study includes six different stations, the values are presented in ranges.**

"However, there were instances where the predicted data surpassed the upper limit of the 95% confidence interval, indicating that the model underestimated the higher values by more than expected. Specifically, 3% of the higher values in the Jakarta station to 5% in the Singapore station exceeded the upper limit of the 95% confidence interval, demonstrating this underestimation. Conversely, there were also cases where the predicted data was below the lower boundary of the 95% confidence interval, suggesting that the model overestimated the very low stable isotope contents. In these cases, 4% of the lower values in the Kuala Lumpur station to 7% of the lower values in the Jayapura station were below the lower boundary, indicating overestimation."

These revisions aim to enhance the clarity and accuracy of our discussion, offering readers a more precise understanding of our model's predictive capabilities and limitations. Please check the revised manuscript.

*Line 335 – again, phrasing is both vague and bold: The stable isotope composition of "precipitation depends mainly on the vapor pressure, precipitation amount, temperature, and potential evaporation" – more accurate to say "predictor variables that were evaluated to have substantial influence on isotope values are ..."*

**Response:**

We apologize for any oversight in our initial draft of the manuscript. In response to your comment, we acknowledge the need for clarity and precision. Therefore, we have revised the relevant sentence in the manuscript to state: "Predictor variables that were evaluated to have a substantial influence on isotope values are vapor pressure, precipitation amount, temperature, and potential evaporation." This revision aims to provide a more accurate and clear explanation.

**technical corrections compact listing of purely technical corrections at the very end ("": typing errors, etc.).**

*In general this manuscript should have been carefully reviewed for typos, grammatircal errors (misplaced commas, conjoined words without a space in between, etc) prior to going out for scientific review.  A partial list of errors is below but I stopped collating these errors as they became numerous:*

**Response:** We apologize for the oversight and acknowledge the numerous typographical and grammatical errors in the manuscript. In response, we have thoroughly reviewed and revised the manuscript to correct these issues. **We engaged a professional editor to ensure that the manuscript adheres to the highest standards of written English, with particular attention to typos, grammatical errors, misplaced commas, and spacing issues.**

We believe these revisions have significantly improved the clarity and readability of the manuscript. We appreciate your patience and understanding and hope that the revised version meets the necessary standards for scientific review.

*Line 16 – contents – should be composition?*

**Response:** Upon reviewing the sentence, we agree that the term "contents" could be replaced with "composition" to more accurately reflect the scientific context of stable isotopes in our study. We have made changes in the revised manuscript.

*Line 26 – missing space*

**Response:**  It has been corrected in the revised manuscript.

*Line 35 – missing "a"*

**Response:**  It has been corrected in the revised manuscript.

*line  99 delete "the" before NOAA*

**Response:**  It has been deleted in the revised manuscript.

*line 157 – should be "training" not train*

**Response:**  It has been corrected to 'training' in the revised manuscript.

*Line 158 – type-o in split*

**Response:**  The typo in 'split' has been corrected in the revised manuscript.

*Line 179 – typo "demonstrated"*

 **Response:**  This typo has been corrected in the revised manuscript.

*Line 220 – is "Lasso" capitalized? It was not earlier in the paper –which ever way is accurate you need to be consistent.*

**Response:**   In our paper, we have chosen to capitalize "Lasso" to adhere to standard conventions and reflect its status as an acronym for "Least Absolute Shrinkage and Selection Operator." We will ensure that this capitalization is maintained consistently throughout the revised manuscript.

*Line 246 – typo – extra "a"*

**Response:**  This typo has been corrected in the revised manuscript.

*Line 305 - typo*

**Response:**  This typo has been corrected in the revised manuscript.

---

## Author Comment (AC7)

Response to Reviewers

Manuscript No.: **hess-2023-299-R1**

Title: **Long-term monthly prediction of precipitation isotopes over Southeast Asia using various machine-learning techniques**

Type of the Manuscript: Article

Corresponding Author: **Dr. Liu Zhongfang**

All authors: Dr. Mojtaba Heydarizad, Dr. Liu Zhongfang,

Dr. Nathsuda Pumijumnong, Dr. Masoud Minaei, Dr. Pouya Salari, Dr. Rogert Sori, and Dr. Hamid Ghalibaf Mohammadabadi

*Dear Editor-in-chief of the Hydrology and Earth System Sciences journal*

Thank you so much for the first round of review of our manuscript. Here **the** manuscript has been revised completely based on the comments of the communities and reviewers. Please find the comments in red **italics**, and **the** written response to the comments in black. (Please consider that important items are shown in **bold**).

**Reviewer#2**

*The manuscript (hess-2023-299) compares the performance of a bunch of machine learning models in simulating the variation of precipitation stable isotope composition using monthly precipitation stable isotope records from six GNIP stations from SE Asia. The application of machine learning methods in hydrological modelling is a rapidly developing research direction. This is also true for the modelling of precipitation stable isotope compositions. Thus the work is timely and of interest. However, the manuscript still needs considerable revision to reach publication. One of my main problems with the manuscript is the lack of a scientific discussion. Section 4 in the current stage hardly goes beyond the description of the results. The other critical issue is the illustration material. Most of the figures and tables needs additional careful editing.*

**Response:** Firstly, We (authors) appreciate the time and effort you have invested in reviewing our work and providing constructive comments. We understand the importance of a robust scientific discussion and the clarity of illustration material in enhancing the quality of our manuscript. Please find below a point-by-point response to your comments:

**1. Scientific Discussion:**

We acknowledge the need for a more substantive scientific discussion in Section 4. In the revised manuscript, we addressed this by:

• Contextualizing Results: We presented our findings within the broader context of hydrological modeling, discussing how our results contribute to the current understanding of precipitation stable isotope variations.

• Methodological Justification: A deeper explanation of the choice of machine learning models has been provided, including their strengths and limitations in the context of our study objectives.

• Inter-model Comparisons: We included a nuanced discussion comparing the performance of different machine learning models, supported by statistical analyses.

• Implications for Hydrology: The implications of our findings for the field of hydrology, particularly in Southeast Asia, has been elaborated upon, discussing potential impacts on future research and practice.

**2. Illustration Material:**

We take the comments on our figures and tables seriously and have made the following enhancements:

• Figure Revisions: Each figure has been critically assessed for its ability to convey the necessary information effectively. We have improve the visual quality and ensure that each figure is accompanied by a comprehensive legend.

• Table Improvements: Tables have been reformatted for better readability and consistency. We have ensured that the data presented is accurate and that the tables are well-integrated with the manuscript's narrative.

• Supplementary Material: Where necessary, we added supplementary material to provide additional clarity on the methodologies used and the results obtained.

We addressed these issues thoroughly and revised the manuscript accordingly. We hope that these changes significantly improve the manuscript and hope that it meets the journal's standards for publication.

**General comments**

*source of the meteorological data: The manuscript vaguely refers to NOAA web site as the source for evaporation and wind speed data in line 139. It is not acceptable since the reader is completely blind which database was used. The actual source should be cited not the web server via the data were accessed. In the next sentence (lines 140-141) it is written that vapor pressure, precipitation amount and air temperature were used from the GNIP. It should be advised to retrieve all meteorological variables from the same source, for instance to avoid resolution problems. In addition, I strongly suggest not using meteo data from the GNIP. Please keep in mind that GNIP is an archive of precipitation isotope data and not for meteorological data. If meteorological data are corrected for measurement inhomogeneity by the national meteo services or agencies it is not transferred to the GNIP. I have my own experience of this.*

**Response:** Thank you for your constructive feedback. We also agree that the specificity of data sources and consistency across datasets are crucial for the integrity of our research.

In your comment, you pointed out the need for a precise citation of the NOAA dataset used for evaporation and wind speed data, rather than a general reference to the NOAA website. **We have provided a complete and accurate reference in our revised manuscript.**

Furthermore, we concur with your recommendation to use a single source for all meteorological variables to avoid issues of resolution and measurement discrepancies. As such, we have revised our methodology to ensure that all meteorological data, including vapor pressure, precipitation amount, and air temperature, are obtained from the same source. This change brings uniformity to our data and strengthens the validity of our analysis.

**Regarding the use of GNIP data for meteorological variables, we appreciate your advice and have decided to refrain from using it,** given that GNIP is primarily an archive of precipitation isotope data. We understand that any corrections made by national meteorological services are not transferred to GNIP, which could lead to inaccuracies in our study. Therefore, we have modified our study and now rely solely on the **NOAA datasets** for our meteorological data needs in the revised manuscript.

We are confident that these revisions address your concerns and significantly improve the manuscript.

**Structural problems:**

- *The methodological description from line 218 to 232 should be moved to Section 3.*

**Response:** In response to your comment, we have carefully reviewed the structure of our manuscript and agree that the section in question would be better placed within the methodology section. This relocation provides a more logical sequence, allowing readers to understand the methods before delving into the results and discussion.

We have made the necessary adjustments in the revised manuscript, ensuring that the transition is smooth and the content in Section 3 is coherent and comprehensive. Please check the revised manuscript.

- *If I understand well, Authors consider VAR as the "gold standard" in the forecasting exercise and rank the ML predictions according to their accuracy compared to the VAR forecasts. However, it is not clear from the text why the VAR forecast can serve as a reference. Instead a year commonly covered each of the six station records could be retained from the ML training and could be used as a reference to compare the performance of the models.*

**Response:** Regarding your comment lets clarify our methodology and the rationale behind our choices step by step.

**Regarding the Use of VAR as a Reference:**

We acknowledge your suggestion to use a year of data from each station as a reference for model comparison. However, we would like to clarify the design of our study, which encompasses two distinct parts: simulation and forecasting.

**Simulation Phase:**

In the simulation phase, we employed machine learning (ML) techniques to simulate the stable isotope content in precipitation across six GNIP stations in Southeast Asia. For this phase, we used the entire

dataset available for each station, except for the last year, which was reserved for (forecasting section). We evaluated the performance of various ML models using established evaluation metrics to determine the most accurate models for each station.

**Forecasting Phase:**

In the forecasting phase, we aimed to forecast the stable isotope content of precipitation using the most accurate ML models identified during the simulation phase, as well as using the Vector Autoregression (VAR) method. The accuracy of these forecasting models was then evaluated using the last year of measured isotope data, which had been set aside from the initial dataset.

**Difference Between Simulating and Forecasting:**

Simulating and forecasting are two distinct tasks with different objectives and methodologies. Simulating involves generating synthetic data based on existing patterns and relationships within the dataset. This is typically done to understand the system's behavior under various scenarios. Forecasting, on the other hand, aims to predict future values based on historical data. In the context of our study, forecasting using ML models involves training the models on historical data (excluding the last year used for the forecasting part) to capture underlying patterns and then using these trained models to forecast the stable isotope content for the reserved future period.

**Why We Kept the Last Year of Sampling:**

We kept the last year of sampling in each station separate because we needed to use the rest of the dataset, from the first year until the year before the last year, for training and testing the simulation ML models. The last year was reserved for evaluating our forecasting models. When developing forecasting models, it is essential to evaluate them with actual measured data to determine their accuracy. By using the last year of sampling data, which was excluded from our dataset for the simulation phase, we were able to compare the forecasted values with the measured values to evaluate the performance of both the ML and VAR models. This approach allowed us to determine which model, ML or VAR, was better for forecasting.

**Why VAR is Used as a Benchmark:**

We chose VAR as a benchmark for several reasons. VAR is a widely recognized and accepted method for time series forecasting, known for its robust performance in various applications. By using VAR, we

intended to provide a standard reference point to evaluate the forecasting capabilities of ML models. This comparison allowed us to determine whether ML models could match or surpass the accuracy of a traditional, well-established forecasting method.

**Importance of Our Approach:**

**Our study is pioneering in that it simultaneously investigates the simulation and forecasting capabilities of ML models for stable isotope content in precipitation. To our knowledge, no previous study has undertaken this dual approach. By doing so, we aimed to provide a comprehensive analysis of ML techniques, assessing their accuracy in both simulating historical data and forecasting future data. This dual assessment fills a significant gap in the literature and offers valuable insights into the broader applicability of ML models in environmental sciences.**

**Conclusion:**

To summarize, we used VAR as a reference because it is a common and accepted model for forecasting time series. Our approach aimed to explore the effectiveness of ML models not only in simulation but also in forecasting, providing a holistic view of their capabilities. We believe this approach enhances the robustness and relevance of our findings.

*The annotations are unreadable in Figs 4, 5, 6, and 7. I suggest changing the layout from the current 2×3 to 3×2 in Figs 4, 5 and 6. It will allow the authors to increase the panels. In addition, I strongly suggest increasing the font size in each panels. In addition, in the legend of Fig 1 "Bangkog" should be corrected to "Bangkok" and "Kota Bahura" should be corrected to "Kota Bharu"*

**Response:** Comment is accepted. We apologize for the oversight and appreciate your suggestions for improving their readability. We made the following changes to address your concerns:

• Figures 4, 5, and 6: The layout has been adjusted from the current 2×3 to a 3×2 format. This change allows for an increase in the size of the panels, making the annotations more legible.

• Font Size: we increased the font size in the annotations of Figs 4, 5, 6, and 7 to ensure that they are easily readable.

• Figure 1 Legend Corrections: The misspellings in the legend of Fig 1 has been corrected to "Bangkok" and "Kota Bharu" to accurately reflect the names of the locations.

These changes have been implemented in the revised version of the manuscript to enhance clarity and accuracy. Please check the revised manuscript.

**Specific comments:**

*line 26: insert a space after the full stop.*

**Response:** Comment is accepted. The formatting error on line 26 has been corrected.

*line 27: No need to introduce the abbreviation "(VAR)" here since it is not used elsewhere in the abstract.*

**Response:** The reviewer's comment has been acknowledged and the abbreviation "(VAR)" has been removed from line 27 of the abstract. Please check the revised manuscript.

*line 39: Beside (or instead of) the classic Clark & Fritz book, a more recent review should be cited e.g. Bowen et al., 2019 (https://doi.org/10.1146/annurev-earth-053018-060220 )*

**Response:** Comment is accepted. We have revised the manuscript to include the reference to Bowen et al. (2019) alongside the classic Clark & Fritz reference, as well as several additional recent references. This provides a more comprehensive and up-to-date overview of the topic. Please check the revised manuscript.

*lines 42 and 44 I suggest moving the citations "IAEA/GNIP 2018" from kine 44 to the end of the sentence in line 42 and citing the most recent review from the IAEA Hydrology group in line 44: Vystavyna et al., 2021 (https://doi.org/10.1038/s41598-021-98094-6 )*

**Response:** Comment is accepted. We have moved the citations "IAEA/GNIP 2018" from line 44 to the end of the sentence in line 42. Additionally, we have included the most recent review from the IAEA Hydrology group as recommended in line 44. Please check the revised manuscript.

*line 79: I suggest replacing "Am" with "monsoon climate"*

**Response:** The reviewer's suggestion has been implemented. The term "Am" has been replaced with "monsoon climate" on line 79 to provide clearer context within the manuscript. Please check the revised manuscript.

*line 128: Please correct the superscript formats in the equation. In addition, 1 should be deleted from the exponent.*

**Response:** The reviewer's comment regarding the equation on line 128 has been addressed. The superscript formatting has been corrected, and the extraneous '1' in the exponent has been removed to accurately reflect the intended mathematical expression. Please check the revised manuscript.

*line 131: The sentence needs revision. The mentioned analytical uncertainties surely refers to the delta values rather than the heavy isotopes.*

**Response:** We have revised the sentence on line 131 to accurately reflect that the uncertainties pertain to the delta values of the isotopes, not the isotopes themselves. Please check the revised manuscript.

*line 135 (and also elsewhere): "potential air evaporation" sounds strange. Probably "air" should be omitted?*

**Response:** Comment is accepted, and the term "air" has been removed from "potential air evaporation" for clarity. We also checked and corrected this term in other parts of the manuscript.

*line 149: "M.H." seems to be a mistake in the citation.*

**Response:** Upon review, we found that "M.H." was a mistake. We have corrected it in the revised manuscript.

*line 158: "spilited" should be changed to "splitted"*

**Response:** Comment is accepted. We have corrected this typo and other similar errors in the revised manuscript. Please review the revised version.

*line 210: It is unclear from the current text which results are referred at the beginning of the sentence.*

**Response:** We clarified in the manuscript that the results referred to at the beginning of the sentence on line 210 **are those obtained by Pearson correlation coefficient in the previous paragraph.** This modification aims to prevent further misunderstanding and vagueness. Please review the revised version of the manuscript for enhanced clarity and coherence.

*line 251: Please check the text. Is it possible that you meant to write "fairly weak" instead of "fairly strong"?*

**Response:** We also agree that "fairly weak" is indeed the correct term to describe the relationship being discussed. The initial use of "fairly strong" was a misrepresentation of the observed correlation between the variables in our study.

To provide further clarification, our intention was to highlight the nuanced influence of temperature on stable isotope composition in tropical regions, as compared to non-tropical regions. In tropical regions, such as those examined in our study, the amount of precipitation plays a significant role due to the influence of seasonal monsoon patterns. This factor often supersedes the impact of air temperature on stable isotope composition. On the other hand, in non-tropical regions, where temperature fluctuations are more pronounced and consistent throughout the year, temperature stands out as a dominant factor in determining $\delta^{18}O$ values, alongside the amount of precipitation.

We have carefully amended the manuscript to reflect this distinction and ensure that our findings are presented with accuracy. Please check the revised manuscript.

**Suggestions on Table 1 and Table 2**

**The layout of both tables could be improved.**

*Table 1: If you introduce the abbreviation LR for Lasso regression in the table title than you can use it in the table which will help the readability of the table. In addition, "$\delta^{18}O$ (VSMOW‰)" and $\delta^2H$ (VSMOW‰)" should go to the header of the first and second part of the table, respectively, to eliminate the current "Isotope" column, which again could help to make this table more compact and readable.*

**Response:** We accept your suggestions regarding the formatting of Table 1. We have introduced the abbreviation LR for Lasso Regression in the table title and have used this abbreviation consistently within the table to improve readability. Additionally, we have moved $\delta^{18}O$ (VSMOW‰) and $\delta^2H$ (VSMOW‰) to the headers of the respective parts of the table, eliminating the need for a separate "Isotope" column. These changes have made the table more compact and easier to read.

*Table 2: Similar suggestion as above. "$\delta^{18}O$" and $\delta^2H$" should go to the header above the methods to eliminate the current "Isotope" columns, to make this table more compact and readable.*

**Response:** Comment is accepted. We also agree that moving the "$\delta^{18}O$" and "$\delta^2H$" to the header above the methods and removing the current "Isotope" columns would streamline the table, making it more compact and easier to read. We have conducted all these changes to enhance the clarity of the data presented. Please check the revised manuscript.